# MULTIMODAL LEARNING WITHOUT LABELED MULTIMODAL DATA: GUARANTEES AND APPLICATIONS

**Paul Pu Liang**[1], **Chun Kai Ling**[2], **Yun Cheng**[3], **Alex Obolenskiy**[1], **Yudong Liu**[1],
**Rohan Pandey**[1], **Alex Wilf**[1], **Louis-Philippe Morency**[1], **Ruslan Salakhutdinov**[1]
[1]Carnegie Mellon University, [2]Columbia University, [3]Princeton University
pliang@cs.cmu.edu, cl4488@columbia.edu

## ABSTRACT

In many machine learning systems that jointly learn from multiple modalities, a core research question is to understand the nature of *multimodal interactions*: how modalities combine to provide new task-relevant information that was not present in either alone. We study this challenge of interaction quantification in a semi-supervised setting with only labeled unimodal data and naturally co-occurring multimodal data (e.g., unlabeled images and captions, video and corresponding audio) but when labeling them is time-consuming. Using a precise information-theoretic definition of interactions, our key contribution is the derivation of lower and upper bounds to quantify the amount of multimodal interactions in this semi-supervised setting. We propose two lower bounds: one based on the *shared information* between modalities and the other based on *disagreement* between separately trained unimodal classifiers, and derive an upper bound through connections to approximate algorithms for *min-entropy couplings*. We validate these estimated bounds and show how they accurately track true interactions. Finally, we show how these theoretical results can be used to estimate multimodal model performance, guide data collection, and select appropriate multimodal models for various tasks.

## 1 INTRODUCTION

A core research question in multimodal learning is to understand the nature of *interactions* between modalities for a task: how much information is shared between both modalities, lies in each modality alone, and the emergence of new task-relevant information during learning from both modalities that was not present in either modality alone (Liang et al., 2022b). In settings where labeled multimodal data is abundant, the study of multimodal interactions in the supervised setting has inspired fundamental advances in theoretical analysis (Hessel & Lee, 2020; Liang et al., 2023a; Sridharan & Kakade, 2008), representation learning (Jayakumar et al., 2020; Radford et al., 2021), and the selection of suitable multimodal models for various real-world tasks (Liang et al., 2023a).

In this paper, we study the problem of interaction quantification in a setting where there is only *unlabeled multimodal data* $\mathcal{D}_M = \{(x_1, x_2)\}$ and some *labeled unimodal data* $\mathcal{D}_i = \{(x_i, y)\}$ collected separately for each modality. This multimodal semi-supervised paradigm is reminiscent of many real-world settings with separate unimodal datasets like visual recognition (Deng et al., 2009) and text classification (Wang et al., 2018), as well as naturally co-occurring multimodal data (e.g., news images and captions or video and audio), but when labeling them is time-consuming (Hsu et al., 2018; Hu et al., 2019) or impossible due to partially observed modalities (Liang et al., 2022a) or privacy concerns (Che et al., 2023). Despite these data constraints, we still want to understand how the modalities can share, exchange, and create information in order to inform our decisions on data collection and modeling (Jayakumar et al., 2020; Liang et al., 2023a; Zadeh et al., 2017).

Using a precise information-theoretic definition of interactions (Bertschinger et al., 2014), our key contributions are the derivations of lower and upper bounds to quantify multimodal interactions in this semi-supervised setting with only $\mathcal{D}_i$ and $\mathcal{D}_M$. We propose two lower bounds: the first relates interactions with the amount of *shared information* between modalities, and the second is based on the *disagreement* of classifiers trained separately on each modality. Finally, we propose an upper bound through connections to approximate algorithms for *min-entropy couplings* (Cicalese & Vaccaro, 2002). To validate our bounds, we experiment on both synthetic and large real-world datasets with varying amounts of interactions. In addition, these theoretical results naturally yield new guarantees regarding the performance of multimodal models. By analyzing the relationship between interaction estimates and downstream task performance assuming optimal multimodal classifiers are

trained on labeled multimodal data, we can *closely predict multimodal model performance, before even training the model itself.* These performance estimates also help develop new guidelines for deciding when to *collect additional modality data* and *select the appropriate multimodal fusion models.* We believe these results shed light on the intriguing connections between multimodal interactions, modality disagreement, and model performance, and release our code and models at https://github.com/pliang279/PID.

## 2  RELATED WORK AND TECHNICAL BACKGROUND

### 2.1  SEMI-SUPERVISED MULTIMODAL LEARNING

Let $\mathcal{X}_i$ and $\mathcal{Y}$ be finite sample spaces for features and labels. Define $\Delta$ to be the set of joint distributions over $(\mathcal{X}_1, \mathcal{X}_2, \mathcal{Y})$. We are concerned with features $X_1, X_2$ (with support $\mathcal{X}_i$) and labels $Y$ (with support $\mathcal{Y}$) drawn from some distribution $p \in \Delta$. We denote the probability mass function by $p(x_1, x_2, y)$, where omitted parameters imply marginalization. Many real-world applications such as multimedia and healthcare naturally exhibit multimodal data (e.g., images and captions, video and audio, multimodal medical readings) which are difficult to label (Liang et al., 2022a; Radford et al., 2021; Singh et al., 2022; Yu & Liu, 2004; Zellers et al., 2022). As such, rather than the full distribution from $p$, we only have partial datasets:

- *Labeled unimodal* data $\mathcal{D}_1 = \{(x_1, y) : \mathcal{X}_1 \times \mathcal{Y}\}$, $\mathcal{D}_2 = \{(x_2, y) : \mathcal{X}_2 \times \mathcal{Y}\}$.
- *Unlabeled multimodal* data $\mathcal{D}_M = \{(x_1, x_2) : \mathcal{X}_1 \times \mathcal{X}_2\}$.

$\mathcal{D}_1, \mathcal{D}_2$ and $\mathcal{D}_M$ follow the *pairwise marginals* $p(x_1, y), p(x_2, y)$ and $p(x_1, x_2)$. We define $\Delta_{p_{1,2}} = \{q \in \Delta : q(x_i, y) = p(x_i, y) \ \forall y \in \mathcal{Y}, x_i \in \mathcal{X}_i, i \in [2]\}$ as the set of joint distributions which agree with the labeled unimodal data $\mathcal{D}_1$ and $\mathcal{D}_2$, and $\Delta_{p_{1,2,12}} = \{r \in \Delta : r(x_1, x_2) = p(x_1, x_2), r(x_i, y) = p(x_i, y)\}$ as the set of joint distributions which agree with all $\mathcal{D}_1, \mathcal{D}_2$ and $\mathcal{D}_M$.

### 2.2  MULTIMODAL INTERACTIONS AND INFORMATION THEORY

The study of **multimodal interactions** aims to quantify the information shared between both modalities, in each modality alone, and how modalities can combine to form new information not present in either modality, eventually using these insights to design machine learning models to capture interactions from large-scale multimodal datasets (Liang et al., 2022b). Existing literature has primarily studied the interactions captured by trained models, such as using Shapley values (Ittner et al., 2021) and Integrated gradients (Sundararajan et al., 2017; Tsang et al., 2018; Liang et al., 2023b) to measure the importance a model assigns to each modality, or approximating trained models with additive or non-additive functions to determine what functions are best suited to capture interactions (Friedman & Popescu, 2008; Sorokina et al., 2008; Hessel & Lee, 2020). However, these measure interactions captured by a trained model - *our work is fundamentally different in that interactions are properties of data.* Quantifying the interactions in data, independent of trained models, allows us to characterize datasets, predict model performance, and perform model selection, prior to choosing and training a model altogether. Prior work in understanding data interactions to design multimodal models is often driven by intuition, such as using contrastive learning (Poklukar et al., 2022; Radford et al., 2021; Tosh et al., 2021), correlation analysis (Andrew et al., 2013), and agreement (Ding et al., 2022) for shared information (e.g., images and descriptive captions), or using tensors and multiplicative interactions (Zadeh et al., 2017; Jayakumar et al., 2020) for higher-order interactions (e.g., in expressions of sarcasm from speech and gestures).

To fill the gap in data quantification, **information theory** has emerged as a theoretical foundation since it naturally formalizes information and its sharing as statistical properties of data distributions. Information theory studies the information that one random variable ($X_1$) provides about another ($X_2$), as quantified by Shannon's mutual information (MI) and conditional MI:

$$I(X_1; X_2) = \int p(x_1, x_2) \log \frac{p(x_1, x_2)}{p(x_1)p(x_2)} d\boldsymbol{x}, \quad I(X_1; X_2|Y) = \int p(x_1, x_2, y) \log \frac{p(x_1, x_2|y)}{p(x_1|y)p(x_2|y)} d\boldsymbol{x} dy.$$

$I(X_1; X_2)$ measures the amount of information (in bits) obtained about $X_1$ by observing $X_2$, and by extension, $I(X_1; X_2|Y)$ is the expected value of MI given the value of a third (e.g., task $Y$).

To generalize information theory for multimodal interactions, Partial information decomposition (PID) (Williams & Beer, 2010) decomposes the total information that two modalities $X_1, X_2$ provide about a task $Y$ into 4 quantities: $I_p(\{X_1, X_2\}; Y) = R + U_1 + U_2 + S$, where $I_p(\{X_1, X_2\}; Y)$ is the MI between the joint random variable $(X_1, X_2)$ and $Y$. These 4 quantities are: redundancy $R$ for

the task-relevant information shared between $X_1$ and $X_2$, uniqueness $U_1$ and $U_2$ for the information present in only $X_1$ or $X_2$ respectively, and synergy $S$ for the emergence of new information only when both $X_1$ and $X_2$ are present (Bertschinger et al., 2014; Griffith & Koch, 2014):

**Definition 1.** *(Multimodal interactions) Given $X_1$, $X_2$, and a target $Y$, we define their redundant (R), unique ($U_1$ and $U_2$), and synergistic (S) interactions as:*

$$R = \max_{q \in \Delta_{p_{1,2}}} I_q(X_1; X_2; Y), \quad U_1 = \min_{q \in \Delta_{p_{1,2}}} I_q(X_1; Y | X_2), \quad U_2 = \min_{q \in \Delta_{p_{1,2}}} I_q(X_2; Y | X_1), \quad (1)$$

$$S = I_p(\{X_1, X_2\}; Y) - \min_{q \in \Delta_{p_{1,2}}} I_q(\{X_1, X_2\}; Y), \quad (2)$$

*where the notation $I_p(\cdot)$ and $I_q(\cdot)$ disambiguates mutual information (MI) under $p$ and $q$ respectively.*

$I(X_1; X_2; Y) = I(X_1; X_2) - I(X_1; X_2 | Y)$ is a multivariate extension of information theory (Bell, 2003; McGill, 1954). Most importantly, $R$, $U_1$, and $U_2$ can be computed exactly using convex programming over distributions $q \in \Delta_{p_{1,2}}$ with access only to the marginals $p(x_1, y)$ and $p(x_2, y)$ by solving a convex optimization problem with linear marginal-matching constraints $q^* = \arg\max_{q \in \Delta_{p_{1,2}}} H_q(Y | X_1, X_2)$ (Bertschinger et al., 2014; Liang et al., 2023a), see Appendix B.2 for more details. This gives us an elegant interpretation that we need only labeled unimodal data in each feature from $\mathcal{D}_1$ and $\mathcal{D}_2$ to estimate redundant and unique interactions. Unfortunately, $S$ is impossible to compute via equation (2) when we do not have access to the full joint distribution $p$, since the first term $I_p(\{X_1, X_2\}; Y)$ is unknown.

It is worth noting that other valid information-theoretic definitions of multimodal interactions also exist, but are known to suffer from issues regarding over- and under-estimation, and may even be negative; these are critical problems with the application of information theory for shared $I(X_1; X_2; Y)$ and unique information $I(X_1; Y | X_2)$, $I(X_2; Y | X_1)$ often quoted in the co-training (Blum & Mitchell, 1998; Balcan et al., 2004) and multi-view learning (Tosh et al., 2021; Tsai et al., 2020; Tian et al., 2020; Sridharan & Kakade, 2008) literature. We refer the reader to Griffith & Koch (2014) for a full discussion. We choose the one in Definition 1 above since it fulfills several desirable properties, but our results can be extended to other definitions as well.

## 3 ESTIMATING SEMI-SUPERVISED MULTIMODAL INTERACTIONS

Our goal is to estimate multimodal interactions $R$, $U_1$, $U_2$, and $S$ assuming access to only semi-supervised multimodal data $\mathcal{D}_1$, $\mathcal{D}_2$, and $\mathcal{D}_M$. Our first insight is that while $S$ cannot be computed exactly, $R$, $U_1$, and $U_2$ can be computed from equation 1 with access to only semi-supervised data. Therefore, studying the relationships between $S$ and other multimodal interactions is key to its estimation. Using these relationships, we will then derive lower and upper bounds for synergy in the form $\underline{S} \leq S \leq \overline{S}$. Crucially, $\underline{S}$ and $\overline{S}$ depend *only* on $\mathcal{D}_1$, $\mathcal{D}_2$, and $\mathcal{D}_M$.

### 3.1 UNDERSTANDING RELATIONSHIPS BETWEEN INTERACTIONS

We start by identifying two important relationships, between $S$ and $R$, and between $S$ and $U$.

**Synergy and redundancy** Our first relationship stems from the case when two modalities contain shared information about the task. In studying these situations, a driving force for estimating $S$ is the amount of shared information $I(X_1; X_2)$ between modalities, with the intuition that more shared information naturally leads to redundancy which gives less opportunity for new synergistic interactions. Mathematically, we formalize this by relating $S$ to $R$,

$$S = R - I_p(X_1; X_2; Y) = R - I_p(X_1; X_2) + I_p(X_1; X_2 | Y). \quad (3)$$

implying that synergy exists when there is high redundancy and low (or even negative) three-way MI $I_p(X_1; X_2; Y)$. By comparing the difference in $X_1$, $X_2$ dependence with and without the task (i.e., $I_p(X_1; X_2)$ vs $I_p(X_1; X_2 | Y)$), 2 cases naturally emerge (see left side of Figure 1):

1. **S > R**: When both modalities do not share a lot of information as measured by low $I(X_1; X_2)$, but conditioning on $Y$ *increases* their dependence: $I(X_1; X_2 | Y) > I(X_1; X_2)$, then there is synergy between modalities when combining them for task $Y$. This setting is reminiscent of common cause structures. Examples of these distributions in the real world are multimodal question answering, where the image and question are less dependent (some questions like 'what is the color of the car' or 'how many people are there' can be asked for many images), but the

Figure 1: We study the relationships between (left) *synergy and redundancy* as a result of the task $Y$ either increasing or decreasing the shared information between $X_1$ and $X_2$ (i.e., common cause structures as opposed to redundancy in common effect), as well as (right) *synergy and uniqueness* due to the disagreement between unimodal predictors resulting in a new prediction $y \neq y_1 \neq y_2$ (rather than uniqueness where $y = y_2 \neq y_1$).

    answer (e.g., 'blue car') connects the two modalities, resulting in dependence given the label. As expected, $S = 4.92, R = 0.79$ for the VQA 2.0 dataset (Goyal et al., 2017).

2. **R > S**: Both modalities share a lot of information but conditioning on $Y$ *reduces* their dependence: $I(X_1; X_2) > I(X_1; X_2|Y)$, which results in more redundant than synergistic information. This setting is reminiscent of common effect structures. A real-world example is in detecting sentiment from multimodal videos, where text and video are highly dependent since they are emitted by the same speaker, but the sentiment label explains away some of the dependencies between both modalities. Indeed, for multimodal sentiment analysis from text, video, and audio of monologue videos on MOSEI (Zadeh et al., 2018), $R = 0.26$ and $S = 0.04$.

**Synergy and uniqueness**    The second relationship arises when two modalities contain disagreeing information about the task, and synergy arises due to this disagreement in information. To illustrate this, suppose $y_1 = \arg\max_y p(y|x_1)$ is the most likely prediction from the first modality, $y_2 = \arg\max_y p(y|x_2)$ for the second modality, and $y = \arg\max_y p(y|x_1, x_2)$ is the true multimodal prediction. There are again 2 cases (see right side of Figure 1):

1. **U > S**: Multimodal prediction $y = \arg\max_y p(y|x_1, x_2)$ is the same as one of the unimodal predictions (e.g., $y = y_2$), in which case unique information in modality 2 leads to the outcome and there is no synergy. A real-world dataset is MIMIC involving mortality and disease prediction from tabular patient data and time-series medical sensors (Johnson et al., 2016) which primarily shows unique information in the tabular modality. The disagreement on MIMIC is high at $0.13$, but since disagreement is due to a lot of unique information, there is less synergy $S = 0.01$.

2. **S > U**: Multimodal prediction $y$ is different from both $y_1$ and $y_2$, then both modalities interact synergistically to give rise to a final outcome different from both disagreeing unimodal predictions. This type of joint distribution is indicative of real-world expressions of sarcasm from language and speech - the presence of sarcasm is typically detected due to a contradiction between what is expressed in language and speech, as we observe from the experiments on MUSTARD (Castro et al., 2019) where $S = 0.44$ and disagreement $= 0.12$ are both large.

## 3.2   Lower and upper bounds on synergy

Given these relationships between synergy and other interactions, we now derive bounds on $S$. We present two lower bounds $\underline{S}_R$ and $\underline{S}_U$, which are based on redundancy and uniqueness, as well as an upper bound $\overline{S}$. We also describe the computational complexity for computing each bound.

*Remark on high dimensional, continuous modalities.* Our theoretical results are concerned with *finite* spaces for features and labels. However, this may be restrictive when working with real-world datasets (e.g., images, video, text) which are often continuous and/or high-dimensional. In such situations, we preprocess by performing discretization of each modality via clustering to estimate $p(x_1, y), p(x_2, y), p(x_1, x_2)$, each with a small, finite support. These are subsequently used for the computation of $\underline{S}_R, \underline{S}_U$ and $\overline{S}$. Discretization is a common way to approximate information theoretic quantities like mutual information (Darbellay & Vajda, 1999; Liang et al., 2023a) and for learning representations over high-dimensional modalities (Oord et al., 2018).

**Lower bound using redundancy**    Our first lower bound uses the relationship between synergy, redundancy, and dependence in equation 3. In semi-supervised settings, we can compute $R$ exactly from $p(x_1, y), p(x_2, y)$, as well as the shared information $I(X_1; X_2)$ from $p(x_1, x_2)$. However, $I_p(X_1; X_2|Y)$ cannot be computed without access to the full distribution $p$. In Theorem 1, we obtain a lower bound on $I_p(X_1; X_2|Y)$, resulting in a lower bound $\underline{S}_R$ for synergy.

**Theorem 1.** *(Lower-bound on synergy via redundancy) We relate $S$ to modality dependence*

$$\underline{S}_R = R - I_p(X_1; X_2) + \min_{r \in \Delta_{p_{1,2,12}}} I_r(X_1; X_2|Y) \leq S \tag{4}$$

We include the full proof in Appendix B.3. This bound compares $S$ to $R$ via the difference of their dependence $I_p(X_1; X_2)$ and their dependence given the task $I_p(X_1; X_2|Y)$. Since the full distribution $p$ is not available to compute $I_p(X_1; X_2|Y)$, we prove a lower bound using conditional MI computed with respect to a set of auxiliary distributions $r \in \Delta_{p_{1,2,12}}$ that are close to $p$, as measured by matching both unimodal marginals $r(x_i, y) = p(x_i, y)$ and modality marginals $r(x_1, x_2) = p(x_1, x_2)$. If conditioning on the task increases the dependence and $I_r(X_1; X_2|Y)$ is large relative to $I_p(X_1; X_2)$ then we obtain a larger value of $\underline{S}_R$, otherwise if conditioning on the task decreases the dependence and $I_r(X_1; X_2|Y)$ is small relative to $I_p(X_1; X_2)$ then we obtain a smaller value of $\underline{S}_R$.

*Computational complexity.* $R$ and $\min_{r \in \Delta_{p_{1,2,12}}} I_r(X_1; X_2|Y)$ are convex optimization problems solvable in polynomial time with off-the-shelf solvers. $I_p(X_1; X_2)$ can be computed directly.

**Lower bound using uniqueness** Our second bound formalizes the relationship between disagreement, uniqueness, and synergy. The key insight is that while labeled multimodal data is unavailable, the output of unimodal classifiers may be compared against each other. Consider unimodal classifiers $f_i : \mathcal{X}_i \to \mathcal{Y}$ and multimodal classifiers $f_M : \mathcal{X}_1 \times \mathcal{X}_2 \to \mathcal{Y}$. Define *modality disagreement* as:

**Definition 2.** *(Modality disagreement) Given $X_1$, $X_2$, and a target $Y$, as well as unimodal classifiers $f_1$ and $f_2$, we define modality disagreement as $\alpha(f_1, f_2) = \mathbb{E}_{p(x_1, x_2)}[d(f_1, f_2)]$ where $d : \mathcal{Y} \times \mathcal{Y} \to \mathbb{R}^{\geq 0}$ is a distance function in label space scoring the disagreement of $f_1$ and $f_2$'s predictions.*

Connecting *modality disagreement* and synergy via Theorem 2 yields a lower bound $\underline{S}_U$:

**Theorem 2.** *(Lower-bound on synergy via uniqueness, informal) We can relate synergy $S$ and uniqueness $U$ to modality disagreement $\alpha(f_1, f_2)$ of optimal unimodal classifiers $f_1, f_2$ as follows:*

$$\underline{S}_U = \alpha(f_1, f_2) \cdot c - \max(U_1, U_2) \leq S \tag{5}$$

*for some constant $c$ depending on the label dimension $|\mathcal{Y}|$ and choice of label distance function $d$.*

Theorem 2 implies that if there is substantial disagreement $\alpha(f_1, f_2)$ between unimodal classifiers, it must be due to the presence of unique or synergistic information. If uniqueness is small, then disagreement must be accounted for by synergy, thereby yielding a lower bound $\underline{S}_U$. Note that the optimality of unimodal classifiers is important: poorly trained unimodal classifiers could show high disagreement but would be uninformative about true interactions. We include the formal version of the theorem based on Bayes' optimality and a full proof in Appendix B.4.

*Computational complexity.* Lower bound $\underline{S}_U$ can also be computed efficiently by estimating $p(y|x_1)$ and $p(y|x_2)$ over modality clusters or training unimodal classifiers $f_\theta(y|x_1)$ and $f_\theta(y|x_2)$. $U_1$ and $U_2$ can be computed using a convex solver in polynomial time.

Hence, the relationships between $S$, $R$, and $U$ yield two lower bounds $\underline{S}_R$ and $\underline{S}_U$. Note that these bounds *always* hold, so we could take $\underline{S} = \max\{\underline{S}_R, \underline{S}_U\}$.

**Upper bound on synergy** By definition, $S = I_p(\{X_1, X_2\}; Y) - R - U_1 - U_2$. However, $I_p(\{X_1, X_2\}; Y)$ cannot be computed exactly without the full distribution $p$. Using the same idea as lower bound 1, we upper bound synergy by *considering the worst-case maximum* $I_r(\{X_1, X_2\}; Y)$ computed over a set of auxiliary distributions $r \in \Delta_{p_{1,2,12}}$ that match both unimodal marginals $r(x_i, y) = p(x_i, y)$ and modality marginals $r(x_1, x_2) = p(x_1, x_2)$:

$$\max_{r \in \Delta_{p_{1,2,12}}} I_r(\{X_1, X_2\}; Y) = \max_{r \in \Delta_{p_{1,2,12}}} \{H_r(X_1, X_2) + H_r(Y) - H_r(X_1, X_2, Y)\} \tag{6}$$

$$= H_p(X_1, X_2) + H_p(Y) - \min_{r \in \Delta_{p_{1,2,12}}} H_r(X_1, X_2, Y), \tag{7}$$

where the second line follows from the definition of $\Delta_{p_{1,2,12}}$. While the first two terms are easy to compute, the third may be difficult, as shown in the following theorem:

**Theorem 3.** *Solving $r^* = \arg\min_{r \in \Delta_{p_{1,2,12}}} H_r(X_1, X_2, Y)$ is NP-hard, even for a fixed $|\mathcal{Y}| \geq 4$.*

Theorem 3 suggests we cannot tractably find a joint distribution which tightly upper bounds synergy when the feature spaces are large. Fortunately, a relaxation of $r \in \Delta_{p_{1,2,12}}$ to $r \in \Delta_{p_{12,y}}$, where $r(x_1, x_2) = p(x_1, x_2)$ and $r(y) = p(y)$, recovers the classic *min-entropy coupling* problem over $(X_1, X_2)$ and $Y$, which is still NP-hard but admits good approximations (Cicalese & Vaccaro, 2002; Cicalese et al., 2017; Kocaoglu et al., 2017; Compton et al., 2023). Our final upper bound $\overline{S}$ is:

**Theorem 4.** *(Upper-bound on synergy)*

$$S \leq H_p(X_1, X_2) + H_p(Y) - \min_{r \in \Delta_{p_{12,y}}} H_r(X_1, X_2, Y) - R - U_1 - U_2 = \overline{S} \qquad (8)$$

Proofs of Theorem 3, 4, and detailed approximation algorithms for min-entropy couplings are included in Appendix B.5 and B.6.

*Computational complexity.* The upper bound $\overline{S}$ can be computed efficiently since solving the variant of the min-entropy problem in Theorem 4 admits approximations that can be computed in time $O(k \log k)$ where $k = \max(|\mathcal{X}_1|, |\mathcal{X}_2|)$. All other entropy and $R, U_1, U_2$ terms are easy to compute (or have been computed via convex optimization from the lower bounds).

Practically, calculating all three bounds is extremely simple, with just a few lines of code. The computation takes < 1 minute and < 180 MB memory space on average for our large datasets ($1,000$-$20,000$ datapoints), more efficient than training even the smallest multimodal prediction model which takes at least 3x time and 15x memory. As a result, *these bounds scale to large and high-dimensional multimodal datasets found in the real world*, which we verify in the following experiments.

## 4 EXPERIMENTS

We design comprehensive experiments to validate these estimated bounds and relationships between different multimodal interactions. Using these results, we describe applications in estimating optimal multimodal performance before training the model itself, which can be used to guide data collection and select appropriate multimodal models for various tasks.

### 4.1 VERIFYING INTERACTION ESTIMATION IN SEMI-SUPERVISED LEARNING

**Synthetic bitwise datasets** Let $\mathcal{X}_1 = \mathcal{X}_2 = \mathcal{Y} = \{0, 1\}$. We generate joint distributions $\Delta$ by sampling $100,000$ vectors from the 8-dim probability simplex and assigning them to $p(x_1, x_2, y)$.

**Large real-world multimodal datasets** We use a collection of 10 real-world datasets from MultiBench (Liang et al., 2021) which add up to a size of more than $700,000$ datapoints.

1. MOSI: $2,199$ videos for sentiment analysis (Zadeh et al., 2016),
2. MOSEI: $23,000$ videos for sentiment and emotion analysis (Zadeh et al., 2018),
3. MUSTARD: 690 videos for sarcasm detection (Castro et al., 2019),
4. UR-FUNNY: a dataset of humor detection from $16,000$ TED talk videos (Hasan et al., 2019),
5. MIMIC: $36,212$ examples predicting patient mortality and diseases from tabular patient data and medical sensors (Johnson et al., 2016),
6. ENRICO: $1,460$ examples classifying mobile user interfaces and screenshots (Leiva et al., 2020).
7. IRFL: $6,697$ images and figurative captions (e.g, 'the car is as fast as a cheetah' describing an image with a fast car in it) (Yosef et al., 2023).
8. NYCaps: $1,820$ New York Yimes cartoon images and humorous captions describing these images (Hessel et al., 2022).
9. VQA: $614,000$ questions and answers about natural images (Antol et al., 2015).
10. ScienceQA: $21,000$ questions and answers about science problems with scientific diagrams (Lu et al., 2022).

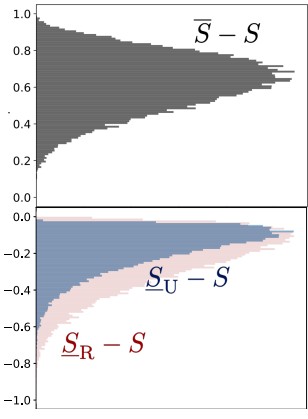

Figure 2: Our two lower bounds $\underline{S}_R$ and $\underline{S}_U$ track actual synergy $S$ from below, and the upper bound $\overline{S}$ tracks $S$ from above. We find that $\underline{S}_R, \underline{S}_U$ tend to approximate $S$ better than $\overline{S}$.

Table 1: We compute lower bounds $\underline{S}_R$, $\underline{S}_U$, and upper bound $\overline{S}$ in semi-supervised multimodal settings and compare them to $S$ assuming knowledge of the full joint distribution $p$. The bounds always hold and track $S$ well on MOSEI, UR-FUNNY, MOSI, and MUSTARD: true $S$ increases as estimated $\underline{S}_R$ and $\underline{S}_U$ increases.

| | MOSEI | UR-FUNNY | MOSI | MUSTARD | MIMIC | ENRICO | NYCAPS | IRFL | VQA | SCIENCEQA |
|---|---|---|---|---|---|---|---|---|---|---|
| $\overline{S}$ | 0.97 | 0.97 | 0.92 | 0.79 | 0.41 | 2.09 | 0.68 | 0.01 | 0.97 | 1.67 |
| $S$ | 0.03 | 0.18 | 0.24 | 0.44 | 0.02 | 1.02 | 0.09 | 0 | 0.05 | 0.16 |
| $\underline{S}_R$ | 0 | 0 | 0.01 | 0.04 | 0 | 0.01 | 0 | 0 | 0 | 0.01 |
| $\underline{S}_U$ | 0.01 | 0.01 | 0.03 | 0.11 | $-0.12$ | $-0.55$ | $-0.03$ | $-0.01$ | 0 | 0 |

| $x_1$ | $x_2$ | $y$ | $p$ |
|---|---|---|---|
| 0 | 0 | 0 | 0 |
| 0 | 0 | 1 | 0.05 |
| 0 | 1 | 0 | 0.03 |
| 0 | 1 | 1 | 0.28 |
| 1 | 0 | 0 | 0.53 |
| 1 | 0 | 1 | 0.03 |
| 1 | 1 | 0 | 0.01 |
| 1 | 1 | 1 | 0.06 |

| $x_1$ | $x_2$ | $y$ | $p$ |
|---|---|---|---|
| 0 | 0 | 0 | 0.25 |
| 0 | 1 | 1 | 0.25 |
| 1 | 0 | 1 | 0.25 |
| 1 | 1 | 0 | 0.25 |

| $x_1$ | $x_2$ | $y$ | $p$ |
|---|---|---|---|
| 0 | 0 | 0 | 0.25 |
| 0 | 1 | 0 | 0.25 |
| 1 | 0 | 1 | 0.25 |
| 1 | 1 | 1 | 0.25 |

| $x_1$ | $x_2$ | $y$ | $p$ |
|---|---|---|---|
| 0 | 0 | 0 | 0.5 |
| 1 | 1 | 1 | 0.5 |

(a) Disagreement XOR  (b) Agreement XOR  (c) $y = x_1$  (d) $y = x_1 = x_2$

Table 2: Four representative examples: (a) disagreement XOR has high disagreement and high synergy, (b) agreement XOR has no disagreement and high synergy, (c) $y = x_1$ has high disagreement and uniqueness but no synergy, and (d) $y = x_1 = x_2$ has high agreement and redundancy but no synergy.

These high-dimensional and continuous modalities require approximating disagreement and mutual information: we train unimodal classifiers $\hat{f}_\theta(y|x_1)$ and $\hat{f}_\theta(y|x_2)$ to estimate disagreement, and we cluster modality features to approximate continuous modalities by discrete distributions with finite support to compute the lower and upper bounds. We summarize the following regarding the validity of each bound (see details in Appendix C):

**1. Overall trends** For the $100,000$ bitwise distributions, we compute $S$, the true value of synergy assuming oracle knowledge of the full multimodal distribution, and compute $\underline{S}_R - S$, $\underline{S}_U - S$, and $S - \overline{S}$ for each point. Plotting these points as a histogram in Figure 2, we find that the two lower bounds track synergy from below ($\underline{S}_R - S$ and $\underline{S}_U - S$ approaching $0$ from below), and the upper bound tracks synergy from above ($S - \overline{S}$ approaching $0$ from above). The two lower bounds are quite tight, as we see that for many points $\underline{S}_R - S$ and $\underline{S}_U - S$ are approaching close to $0$, with an average gap of $0.18$. $\underline{S}_U$ seems to be tighter empirically than $\underline{S}_R$: for half the points, $\underline{S}_U$ is within $0.14$ and $\underline{S}_R$ is within $0.2$ of $S$. For the upper bound, there is an average gap of $0.62$. However, it performs especially well on high synergy data: when $S > 0.6$, the average gap is $0.24$, with more than half of the points within $0.25$ of $S$.

On real-world MultiBench datasets, we show the estimated bounds and actual $S$ computed assuming knowledge of full $p$ in Table 1. The lower and upper bounds track true $S$: as estimated $\underline{S}_R$ and $\underline{S}_U$ increases from MOSEI to UR-FUNNY to MOSI to MUSTARD, true $S$ also increases. For datasets like MIMIC with disagreement but high uniqueness, $\underline{S}_U$ can be negative, but we can rely on $\underline{S}_R$ to give a tight estimate on low synergy. Unfortunately, our bounds do not track synergy well on ENRICO. We believe this is because ENRICO displays all interactions: $R = 0.73, U_1 = 0.38, U_2 = 0.53, S = 0.34$, which makes it difficult to distinguish between $R$ and $S$ using $\underline{S}_R$ or $U$ and $S$ using $\underline{S}_U$ since no interaction dominates over others, and $\overline{S}$ is also quite loose. Given these general observations, we now carefully analyze the relationships between redundancy, uniqueness, and synergy.

**2. Guidelines** We provide a guideline to decide whether a lower or upper bound on synergy can be considered 'close enough'. It is close enough if the maximum interaction can be consistently estimated - often the exact value of synergy is not the most important (e.g, whether $S$ is $0.5$ or $0.6$) but rather synergy relative to other interactions (e.g., if we estimate $S \in [0.2, 0.5]$, and exactly compute $R = U_1 = U_2 = 0.1$, then we know for sure that $S$ is the most important interaction and can collect data or design models based on that). We find that our bounds accurately identify the same highest interaction on all 10 real-world datasets as the true synergy does. Furthermore, we observed that the estimated synergy correlates very well with true synergy: as high as $1.05$ on ENRICO (true $S = 1.02$) and as low as $0.21$ on MIMIC (true $S = 0.02$).

Table 3: Estimated lower, upper, and average bounds on optimal multimodal performance in comparison with the actual best unimodal model, the best simple fusion model, and the best complex fusion model. Our performance estimates closely predict actual model performance, *despite being computed only on semi-supervised data and never training the model itself.*

|  | MOSEI | UR-FUNNY | MOSI | MUSTARD | MIMIC | ENRICO |
|---|---|---|---|---|---|---|
| Estimated upper bound | 1.07 | 1.21 | 1.29 | 1.63 | 1.27 | 0.88 |
| Best complex multimodal | 0.88 | 0.77 | 0.86 | 0.79 | 0.92 | 0.51 |
| Best simple multimodal | 0.85 | 0.76 | 0.84 | 0.74 | 0.92 | 0.49 |
| Best unimodal | 0.82 | 0.74 | 0.83 | 0.74 | 0.92 | 0.47 |
| Estimated lower bound | 0.52 | 0.58 | 0.62 | 0.78 | 0.76 | 0.48 |
| Estimated average | 0.80 | 0.90 | 0.96 | 1.21 | 1.02 | 0.68 |

**3. The relationship between $S$ and $R$** In Table 2b we show the classic AGREEMENT XOR distribution where $X_1$ and $X_2$ are independent, but $Y = 1$ sets $X_1 \neq X_2$ to increase their dependence. $I(X_1; X_2; Y)$ is negative, and $\underline{S}_R = 1 \leq 1 = S$ is tight. On the other hand, Table 2d is an extreme example where the probability mass is distributed uniformly only when $y = x_1 = x_2$ and 0 elsewhere. As a result, $X_1$ is always equal to $X_2$ (perfect dependence), and yet $Y$ perfectly explains away the dependence between $X_1$ and $X_2$ so $I(X_1; X_2|Y) = 0$: $\underline{S}_R = 0 \leq 0 = S$. A real-world example is multimodal sentiment analysis from text, video, and audio on MOSEI, $R = 0.26$ and $S = 0.03$, and as expected the lower bound is small $\underline{S}_R = 0 \leq 0.03 = S$ (Table 1).

**4. The relationship between $S$ and $U$** In Table 2a we show an example called DISAGREEMENT XOR. There is maximum disagreement between $p(y|x_1)$ and $p(y|x_2)$: the likelihood for $y$ is high when $y$ is the opposite bit as $x_1$, but reversed for $x_2$. Given both $x_1$ and $x_2$: $y$ takes a 'disagreement' XOR of the individual marginals, i.e. $p(y|x_1, x_2) = \arg\max_y p(y|x_1)$ XOR $\arg\max_y p(y|x_2)$, which indicates synergy (note that an exact XOR would imply perfect agreement and high synergy). The actual disagreement is $0.15$, $S$ is $0.16$, and $U$ is $0.02$, indicating a very strong lower bound $\underline{S}_U = 0.14 \leq 0.16 = S$. A real-world equivalent dataset is MUSTARD, where the presence of sarcasm is often due to a contradiction between what is expressed in language and speech, so disagreement $\alpha = 0.12$ is the highest out of all the video datasets, giving a lower bound $\underline{S}_U = 0.11 \leq 0.44 = S$.

The lower bound is low when all disagreement is explained by uniqueness (e.g., $y = x_1$, Table 2c), which results in $\underline{S}_U = 0 \leq 0 = S$ ($\alpha$ and $U$ cancel each other out). A real-world equivalent is MIMIC: from Table 1, disagreement is high $\alpha = 0.13$ due to unique information $U_1 = 0.25$, so the lower bound informs us about the lack of synergy $\underline{S}_U = -0.12 \leq 0.02 = S$. Finally, the lower bound is loose when there is synergy without disagreement, such as AGREEMENT XOR ($y = x_1$ XOR $x_2$, Table 2b) where the marginals $p(y|x_i)$ are both uniform, but there is full synergy: $\underline{S}_U = 0 \leq 1 = S$. Real-world datasets include UR-FUNNY where there is low disagreement in predicting humor $\alpha = 0.03$, and relatively high synergy $S = 0.18$, which results in a loose lower bound $\underline{S}_U = 0.01 \leq 0.18 = S$.

**5. On upper bounds for synergy** The upper bound for MUSTARD is close to real synergy, $\overline{S} = 0.79 \geq 0.44 = S$. On MIMIC, the upper bound is the lowest $\overline{S} = 0.41$, matching the lowest $S = 0.02$. Some of the other examples in Table 1 show weaker bounds. This could be because (i) there exists high synergy distributions that match $\mathcal{D}_i$ and $\mathcal{D}_M$, but these are rare in the real world, or (ii) our approximation used in Theorem 4 is loose. We leave these as directions for future work.

**Additional results** In Appendix C and E, we also study the effect of imperfect unimodal predictors and disagreement measurements on our derived bounds, by perturbing the label by various noise levels (from no noise to very noisy) and examining the changes in estimated upper and lower bounds. We found these bounds are quite robust to label noise, still giving close trends of $S$. We also include more discussions studying the relationships between various interactions, and how the relationship between disagreement and synergy can inspire new self-supervised learning methods.

### 4.2 IMPLICATIONS TOWARDS PERFORMANCE, DATA COLLECTION, MODEL SELECTION

Now that we have validated the accuracy of these bounds, we apply them to estimate multimodal performance in semi-supervised settings. This serves as a strong signal for deciding (1) whether to collect paired and labeled data from a second modality, and (2) what type of multimodal fusion method should be used. To estimate performance given $\mathcal{D}_1$, $\mathcal{D}_2$, and $\mathcal{D}_M$, we first compute our lower and upper bounds $\underline{S}$ and $\overline{S}$. Combined with the exact computation of $R$ and $U$, we obtain the total information $I_p(\{X_1, X_2\}; Y)$, and combine a result from Feder & Merhav (1994) with Fano's inequality (Fano, 1968) to yield tight bounds of performance as a function of total information.

**Theorem 5.** *Let* $P_{acc}(f_M^*) = \mathbb{E}_p \left[ \mathbf{1} \left[ f_M^*(x_1, x_2) = y \right] \right]$ *denote the accuracy of the Bayes' optimal multimodal model* $f_M^*$ *(i.e.,* $P_{acc}(f_M^*) \geq P_{acc}(f_M')$ *for all* $f_M' \in \mathcal{F}_M$*). We have that*

$$2^{I_p(\{X_1, X_2\}; Y) - H(Y)} \leq P_{acc}(f_M^*) \leq \frac{I_p(\{X_1, X_2\}; Y) + 1}{\log |\mathcal{Y}|}, \tag{9}$$

*and we can plug in* $R + U_1, U_2 + \underline{S} \leq I_p(\{X_1, X_2\}; Y) \leq R + U_1, U_2 + \overline{S}$ *to obtain lower* $\underline{P}_{acc}(f_M^*)$ *and upper* $\overline{P}_{acc}(f_M^*)$ *bounds on optimal multimodal performance.*

We show the proof in Appendix D. Finally, we summarize estimated multimodal performance as the average $\hat{P}_M = (\underline{P}_{\text{acc}}(f_M^*) + \overline{P}_{\text{acc}}(f_M^*))/2$. A high $\hat{P}_M$ suggests the presence of important joint information from both modalities (not present in each) which could boost accuracy, so it is worthwhile to collect the full distribution $p$ and explore multimodal fusion.

**Setup** For each MultiBench dataset, we implement a suite of unimodal and multimodel models spanning simple and complex fusion. Unimodal models are trained and evaluated separately on each modality. Simple fusion includes ensembling by taking an additive or majority vote between unimodal models (Hastie & Tibshirani, 1987). Complex fusion is designed to learn higher-order interactions as exemplified by bilinear pooling (Fukui et al., 2016), multiplicative interactions (Jayakumar et al., 2020), tensor fusion (Zadeh et al., 2017), and cross-modal self-attention (Tsai et al., 2019). See Appendix D for models and training details. We include unimodal, simple and complex multimodal performance, as well as estimated lower and upper bounds on performance in Table 3.

**RQ1: Estimating multimodal fusion performance** *How well could my multimodal model perform?* We find that estimating interactions enables us to *closely predict multimodal model performance, before even training a model*. For example, on MOSEI, we estimate the performance to be $52\%$ based on the lower bound and $107\%$ based on the upper bound, for an average of $80\%$ which is very close to true model performance ranging from $82\%$ for the best unimodal model, and $85\% - 88\%$ for various multimodal model. Estimated performances for ENRICO, UR-FUNNY, and MOSI are $68\%$, $90\%$, $96\%$, which track true performances $51\%$, $77\%$, $86\%$.

**RQ2: Data collection** *Should I collect multimodal data?* We compare estimated performance $\hat{P}_M$ with the actual difference between unimodal and best multimodal performance in Figure 3 (left). Higher estimated $\hat{P}_M$ correlates with a larger gain from unimodal to multimodal (correlation $\rho = 0.21$ and rises to $0.53$ if ignoring the outlier in MIMIC). MUSTARD and ENRICO show the most opportunity for multimodal modeling. Therefore, a rough guideline is that if the estimated multimodal performance based on semi-supervised data is higher, then collecting the full labeled multimodal data is worth it.

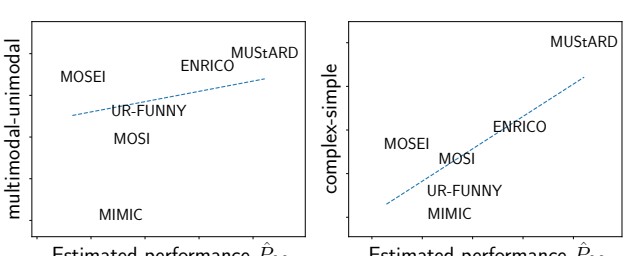

Figure 3: Datasets with higher estimated multimodal performance $\hat{P}_M$ tend to show improvements from unimodal to multimodal (left) and from simple to complex multimodal fusion (right).

**RQ3: Model selection** *What model should I choose for multimodal fusion?* We note strong relationships between estimated performance and the performance of different fusion methods. From Table 3, synergistic datasets like MUSTARD and ENRICO show best multimodal performance only slightly above our estimated lower bound, indicating that there is a lot of room for improvement in better fusion methods. Indeed, more complex fusion methods such as multimodal transformer designed to capture synergy is the best on MUSTARD which matches its high synergy ($72\%$ accuracy). For datasets with less synergy like MOSEI and MIMIC, the best multimodal performance is much higher than the estimated lower bound, indicating that existing fusion methods may already be quite optimal. Indeed, simpler fusion methods such as feature alignment, designed to capture redundancy, are the best on MOSEI which matches its high redundancy ($80\%$ accuracy).

Figure 3 (right) shows a visual comparison, where plotting the performance gap between complex and simple fusion methods against estimated performance $\hat{P}_M$ shows a correlation coefficient of $0.77$. We again observe positive trends between higher estimated performance and improvements with

complex fusion, with large gains on MUSTARD and ENRICO. We expect new methods to further improve the state-of-the-art on these datasets due to their generally high interaction values and low multimodal performance relative to estimated lower bound $\underline{P}_{\text{acc}}(f_M^*)$. Therefore, a rough guideline is that if the estimated multimodal performance based on semi-supervised data is higher, then there is more potential for improvement by trying more complex multimodal fusion strategies.

## 5 CONCLUSION AND BROADER IMPACTS

We proposed estimators of multimodal interactions when observing only *labeled unimodal data* and some *unlabeled multimodal data*, a general semi-supervised setting that encompasses many real-world constraints involving partially observable modalities, limited labels, and privacy concerns. Our key results draw new connections between multimodal interactions, the disagreement of unimodal classifiers, and min-entropy couplings, which yield new insights for estimating multimodal model performance, data analysis, and model selection. We are aware of some potential **limitations**:

1. These estimators only approximate real interactions due to cluster preprocessing or unimodal models, which naturally introduce optimization and generalization errors. We expect progress in density estimators, generative models, and unimodal classifiers to address these problems.
2. It is harder to quantify interactions for certain datasets, such as ENRICO which displays all interactions which makes it difficult to distinguish between $R$ and $S$ or $U$ and $S$.
3. Finally, there exist challenges in quantifying interactions since the data generation process is never known for real-world datasets, so we have to resort to human judgment, other automatic measures, and downstream tasks such as estimating model performance and model selection.

**Future work** should investigate more applications of multivariate information theory in designing self-supervised models, predicting multimodal performance, and other tasks involving feature interactions such as privacy-preserving and fair representation learning from high-dimensional data (Dutta et al., 2020; Hamman & Dutta, 2023). Being able to provide guarantees for fairness and privacy-preserving learning, especially for semi-supervised pretraining datasets, can be particularly impactful.

## ACKNOWLEDGEMENTS

This material is based upon work partially supported by National Science Foundation awards 1722822 and 1750439, National Institutes of Health awards R01MH125740, R01MH132225, R01MH096951 and R21MH130767, and Meta. PPL is supported in part by a Siebel Scholarship and a Waibel Presidential Fellowship. RS is supported in part by ONR grant N000142312368 and DARPA FA87502321015. Any opinions, findings, conclusions, or recommendations expressed in this material are those of the author(s) and do not necessarily reflect the views of the sponsors, and no official endorsement should be inferred. Finally, we would also like to acknowledge feedback from anonymous reviewers who significantly improved the paper and NVIDIA's GPU support.

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

APPENDIX

## A  BROADER IMPACT

Multimodal semi-supervised models are ubiquitous in a range of real-world applications with only labeled unimodal data and naturally co-occurring multimodal data (e.g., unlabeled images and captions, video and corresponding audio) but when labeling them is time-consuming. This paper is our attempt at formalizing the learning setting of multimodal semi-supervised learning, allowing us to derive bounds on the information existing in multimodal semi-supervised datasets and what can be learned by models trained on these datasets. We do not foresee any negative broad impacts of our theoretical results, but we do note the following concerns regarding the potential empirical applications of these theoretical results in real-world multimodal datasets:

**Biases**: We acknowledge risks of potential biases surrounding gender, race, and ethnicity in large-scale multimodal datasets (Bolukbasi et al., 2016), especially those collected in a semi-supervised setting with unlabeled and unfiltered images and captions (Birhane et al., 2021). Formalizing the types of bias in multimodal datasets and mitigating them is an important direction for future work.

**Privacy**: When making predictions from multimodal datasets with recorded human behaviors and medical data, there might be privacy risks of participants. Following best practices in maintaining the privacy and safety of these datasets, (1) these datasets have only been collected from public data that are consented for public release (creative commons license and following fair use guidelines of YouTube) (Castro et al., 2019; Hasan et al., 2019; Zadeh et al., 2018), or collected from hospitals under strict IRB and restricted access guidelines (Johnson et al., 2016), and (2) have been rigorously de-identified in accordance with Health Insurance Portability and Accountability Act such that all possible personal and protected information has been removed from the dataset (Johnson et al., 2016). Finally, we only use these datasets for research purposes and emphasize that any multimodal models trained to perform prediction should only be used for scientific study and should not in any way be used for real-world harm.

## B  DETAILED PROOFS

### B.1  INFORMATION DECOMPOSITION

Partial information decomposition (PID) (Williams & Beer, 2010) decomposes of the total information 2 variables provide about a task $I(\{X_1, X_2\}; Y)$ into 4 quantities: redundancy $R$ between $X_1$ and $X_2$, unique information $U_1$ in $X_1$ and $U_2$ in $X_2$, and synergy $S$. Williams & Beer (2010), who first proposed PIDs, showed that they should satisfy the following consistency equations:

$$R + U_1 = I(X_1; Y), \tag{10}$$
$$R + U_2 = I(X_2; Y), \tag{11}$$
$$U_1 + S = I(X_1; Y | X_2), \tag{12}$$
$$U_2 + S = I(X_2; Y | X_1), \tag{13}$$
$$R - S = I(X_1; X_2; Y). \tag{14}$$

We choose the PID definition by Bertschinger et al. (2014), where redundancy, uniqueness, and synergy are defined by the solution to the following optimization problems:

$$R = \max_{q \in \Delta_p} I_q(X_1; X_2; Y) \tag{15}$$

$$U_1 = \min_{q \in \Delta_p} I_q(X_1; Y | X_2) \tag{16}$$

$$U_2 = \min_{q \in \Delta_p} I_q(X_2; Y | X_1) \tag{17}$$

$$S = I_p(\{X_1, X_2\}; Y) - \min_{q \in \Delta_p} I_q(\{X_1, X_2\}; Y) \tag{18}$$

where $\Delta_p = \{q \in \Delta : q(x_i, y) = p(x_i, y) \ \forall y, x_i, i \in \{1, 2\}\}$, $\Delta$ is the set of all joint distributions over $X_1, X_2, Y$, and the notation $I_p(\cdot)$ and $I_q(\cdot)$ disambiguates MI under joint distributions $p$ and $q$ respectively. The key difference in this definition of PID lies in optimizing $q \in \Delta_p$ to satisfy the marginals $q(x_i, y) = p(x_i, y)$, but relaxing the coupling between $x_1$ and $x_2$: $q(x_1, x_2)$ need not be equal to $p(x_1, x_2)$. The intuition behind this is that one should be able to infer redundancy and uniqueness given only access to separate marginals $p(x_1, y)$ and $p(x_2, y)$, and therefore they should

only depend on $q \in \Delta_p$ which match these marginals. Synergy, however, requires knowing the coupling $p(x_1, x_2)$, and this is reflected in equation (18) depending on the full $p$ distribution.

## B.2 COMPUTING $q^*$, REDUNDANCY, AND UNIQUENESS

According to Bertschinger et al. (2014), it suffices to solve for $q$ using the following max-entropy optimization problem $q^* = \arg \max_{q \in \Delta_p} H_q(Y|X_1, X_2)$, the same $q^*$ equivalently solves any of the remaining problems defined for redundancy, uniqueness, and synergy. This is a concave maximization problem with linear constraints. When $\mathcal{X}_i$ and $\mathcal{Y}$ are small and discrete, we can represent all valid distributions $q(x_1, x_2, y)$ as a set of tensors $Q$ of shape $|\mathcal{X}_1| \times |\mathcal{X}_2| \times |\mathcal{Y}|$ with each entry representing $Q[i, j, k] = p(X_1 = i, X_2 = j, Y = k)$. The problem then boils down to optimizing over valid tensors $Q \in \Delta_p$ that match the marginals $p(x_i, y)$ for the objective function $H_q(Y|X_1, X_2)$. We rewrite conditional entropy as a KL-divergence (Globerson & Jaakkola, 2007), $H_q(Y|X_1, X_2) = \log |\mathcal{Y}| - KL(q\|\tilde{q})$, where $\tilde{q}$ is an auxiliary product density of $q(x_1, x_2) \cdot \frac{1}{|\mathcal{Y}|}$ enforced using linear constraints: $\tilde{q}(x_1, x_2, y) = q(x_1, x_2)/|\mathcal{Y}|$. The KL-divergence objective is recognized as convex, allowing the use of conic solvers such as SCS (O'Donoghue et al., 2016), ECOS (Domahidi et al., 2013), and MOSEK (ApS, 2022).

Finally, optimizing over $Q \in \Delta_p$ that match the marginals can also be enforced through linear constraints: the 3D-tensor $Q$ summed over the second dimension gives $q(x_1, y)$ and summed over the first dimension gives $q(x_2, y)$, yielding the final optimization problem:

$$\arg \max_{Q, \tilde{Q}} KL(Q\|\tilde{Q}), \quad \text{s.t.} \quad \tilde{Q}(x_1, x_2, y) = Q(x_1, x_2)/|\mathcal{Y}|, \tag{19}$$

$$\sum_{x_2} Q = p(x_1, y), \sum_{x_1} Q = p(x_2, y), Q \geq 0, \sum_{x_1, x_2, y} Q = 1. \tag{20}$$

After solving this optimization problem, plugging $q^*$ into (15)-(17) yields the desired estimators for redundancy and uniqueness: $R = I_{q^*}(X_1; X_2; Y), U_1 = I_{q^*}(X_1; Y|X_2), U_2 = I_{q^*}(X_2; Y|X_1)$, and more importantly, can be inferred from access to only labeled unimodal data $p(x_1, y)$ and $p(x_2, y)$. Unfortunately, $S$ is impossible to compute via equation (18) when we do not have access to the full joint distribution $p$, since the first term $I_p(X_1, X_2; Y)$ is unknown. Instead, we will aim to provide lower and upper bounds in the form $\underline{S} \leq S \leq \overline{S}$ so that we can have a minimum and maximum estimate on what synergy could be. Crucially, $\underline{S}$ and $\overline{S}$ should depend *only* on $\mathcal{D}_1, \mathcal{D}_2$, and $\mathcal{D}_M$ in the multimodal semi-supervised setting.

## B.3 LOWER BOUND ON SYNERGY VIA REDUNDANCY (THEOREM 1)

We first restate Theorem 1 from the main text to obtain our first lower bound $\underline{S}_R$ linking synergy to redundancy:

**Theorem 6.** *(Lower-bound on synergy via redundancy, same as Theorem 1) We can relate $S$ to $R$ as follows*

$$\underline{S}_R = R - I_p(X_1; X_2) + \min_{r \in \Delta_{p_{1,2,12}}} I_r(X_1; X_2|Y) \leq S \tag{21}$$

*where $\Delta_{p_{1,2,12}} = \{r \in \Delta : r(x_1, x_2) = p(x_1, x_2), r(x_i, y) = p(x_i, y)\}$. $\min_{r \in \Delta_{p_{1,2,12}}} I_r(X_1; X_2|Y)$ is a max-entropy convex optimization problem which can be solved exactly using linear programming.*

*Proof.* By consistency equation (14), $S = R - I_p(X_1; X_2; Y) = R - I_p(X_1; X_2) + I_p(X_1; X_2|Y)$. Note that $R$ can be computed exactly based on $p(x_1, y), p(x_2, y)$, and $I_p(X_1; X_2)$ can be computed exactly based on $p(x_1, x_2)$, but $I_r(X_1; X_2|Y)$ requires knowledge of the full distribution $p(x_1, x_2, y)$ to compute. We will instead lower bound the conditional mutual information $I_p(X_1, X_2|Y)$ by computing its minimum with respect to a set of auxiliary distributions $r \in \Delta_{p_{1,2,12}}$ that match both unimodal marginals $r(x_i, y) = p(x_i, y)$ and modality marginals $r(x_1, x_2) = p(x_1, x_2)$, which yields a lower bound on synergy. We obtain:

$$\min_{r \in \Delta_{p_{1,2,12}}} I_r(X_1; X_2|Y) = \min_{r \in \Delta_{p_{1,2,12}}} H_r(X_1) - I_r(X_1; Y) - H_r(X_1|X_2, Y) \tag{22}$$

$$= H_p(X_1) - I_p(X_1; Y) - \max_{r \in \Delta_{p_{1,12}}} H_r(X_1|X_2, Y) \tag{23}$$

where in the last line the $p_2$ constraint is removed since $H_r(X_1|X_2, Y)$ is fixed with respect to $p(x_2, y)$. To solve $\max_{r \in \Delta_{p_{1,12}}} H_r(X_1|X_2, Y)$, we observe that it is also a concave maximization problem with linear constraints. When $\mathcal{X}_i$ and $\mathcal{Y}$ are small and discrete, we can represent all valid distributions $r(x_1, x_2, y)$ as a set of tensors $R$ of shape $|\mathcal{X}_1| \times |\mathcal{X}_2| \times |\mathcal{Y}|$ with each entry representing $R[i, j, k] = p(X_1 = i, X_2 = j, Y = k)$. The problem then boils down to optimizing over valid tensors $R \in \Delta_{p_{1,12}}$ that match the marginals $p(x_1, y)$ and $p(x_1, x_2)$. Given a tensor $R$ representing $r$, our objective is the concave function $H_r(X_1|X_2, Y)$ which we rewrite as a KL-divergence $\log |\mathcal{X}_1| - KL(r\|\tilde{r})$ using an auxiliary distribution $\tilde{r} = r(x_2, y) \cdot \frac{1}{|\mathcal{X}_1|}$ and solve it exactly using convex programming with linear constraints:

$$\underset{R, \tilde{R}}{\arg\max} \, KL(R\|\tilde{R}), \quad \text{s.t.} \quad \tilde{R}(x_1, x_2, y) = R(x_2, y)/|\mathcal{Y}|, \tag{24}$$

$$\sum_{x_2} R = p(x_1, y), \sum_y R = p(x_1, x_2), R \ge 0, \sum_{x_1, x_2, y} R = 1. \tag{25}$$

with marginal constraints $R \in \Delta_{p_{1,12}}$ enforced through linear constraints on tensor $R$. Plugging the optimized $r^*$ into (21) yields the desired lower bound $\underline{S}_R = R - I_p(X_1; X_2) + I_{r^*}(X_1; X_2|Y)$. $\qquad \square$

## B.4 LOWER BOUND ON SYNERGY VIA UNIQUENESS (THEOREM 2)

We first restate some notation and definitions from the main text for completeness. The key insight behind Theorem 2, a relationship between synergy and uniqueness, is that while labeled multimodal data is unavailable, the output of unimodal classifiers may be compared against each other. Let $\delta_{\mathcal{Y}} = \{r \in \mathbb{R}_+^{|\mathcal{Y}|} \mid \|r\|_1 = 1\}$ be the probability simplex over labels $\mathcal{Y}$. Consider the set of unimodal classifiers $\mathcal{F}_i \ni f_i : \mathcal{X}_i \to \delta_{\mathcal{Y}}$ and multimodal classifiers $\mathcal{F}_M \ni f_M : \mathcal{X}_1 \times \mathcal{X}_2 \to \delta_{\mathcal{Y}}$.

**Definition 3.** *(Unimodal and multimodal loss) The loss of a given unimodal classifier $f_i \in \mathcal{F}_i$ is given by $L(f_i) = \mathbb{E}_{p(x_i, y)}[\ell(f_i(x_i), y)]$ for a loss function over the label space $\ell : \mathcal{Y} \times \mathcal{Y} \to \mathbb{R}^{\ge 0}$. We denote the same for multimodal classifier $f_M \in \mathcal{F}_M$, with a slight abuse of notation $L(f_M) = \mathbb{E}_{p(x_1, x_2, y)}[\ell(f_M(x_1, x_2), y)]$ for a loss function over the label space $\ell$.*

**Definition 4.** *(Unimodal and multimodal accuracy) The accuracy of a given unimodal classifier $f_i \in \mathcal{F}_i$ is given by $P_{acc}(f_i) = \mathbb{E}_p[\mathbf{1}[f_i(x_i) = y]]$. We denote the same for multimodal classifier $f_M \in \mathcal{F}_M$, with a slight abuse of notation $P_{acc}(f_M) = \mathbb{E}_p[\mathbf{1}[f_M(x_1, x_2) = y]]$.*

An unimodal classifier $f_i^*$ is Bayes-optimal (or simply optimal) with respect to a loss function $L$ if $L(f_i^*) \le L(f_i')$ for all $f_i' \in \mathcal{F}_i$. Similarly, a multimodal classifier $f_M^*$ is optimal with respect to loss $L$ if $L(f_M^*) \le L(f_M')$ for all $f_M' \in \mathcal{F}_M$.

Bayes optimality can also be defined with respect to accuracy, if $P_{acc}(f_i^*) \ge P_{acc}(f_i')$ for all $f_i' \in \mathcal{F}_i$ for unimodal classifiers, or if $P_{acc}(f_M^*) \ge P_{acc}(f_M')$ for all $f_M' \in \mathcal{F}_M$ for multimodal classifiers.

The crux of our method is to establish a connection between *modality disagreement* and a lower bound on synergy.

**Definition 5.** *(Modality disagreement) Given $X_1$, $X_2$, and a target $Y$, as well as unimodal classifiers $f_1$ and $f_2$, we define modality disagreement as $\alpha(f_1, f_2) = \mathbb{E}_{p(x_1, x_2)}[d(f_1, f_2)]$ where $d : \mathcal{Y} \times \mathcal{Y} \to \mathbb{R}^{\ge 0}$ is a distance function in label space scoring the disagreement of $f_1$ and $f_2$'s predictions,*

where the distance function $d$ must satisfy some common distance properties, following Sridharan & Kakade (2008):

**Assumption 1.** *(Relaxed triangle inequality) For the distance function $d : \mathcal{Y} \times \mathcal{Y} \to \mathbb{R}^{\ge 0}$ in label space scoring the disagreement of $f_1$ and $f_2$'s predictions, there exists $c_d \ge 1$ such that*

$$\forall \hat{y}_1, \hat{y}_2, \hat{y}_3 \in \hat{\mathcal{Y}}, \quad d(\hat{y}_1, \hat{y}_2) \le c_d \left( d(\hat{y}_1, \hat{y}_3) + d(\hat{y}_3, \hat{y}_2) \right). \tag{26}$$

**Assumption 2.** *(Inverse Lipschitz condition) For the function $d$, it holds that for all $f$,*

$$\mathbb{E}[d(f(x_1, x_2), f^*(x_1, x_2))] \le |L(f) - L(f^*)| \tag{27}$$

*where $f^*$ is the Bayes optimal multimodal classifier with respect to loss $L$, and*

$$\mathbb{E}[d(f_i(x_i), f_i^*(x_i))] \le |L(f_i) - L(f_i^*)| \tag{28}$$

*where $f_i^*$ is the Bayes optimal unimodal classifier with respect to loss $L$.*

**Assumption 3.** *(Classifier optimality) For any unimodal classifiers $f_1, f_2$ in comparison to the Bayes' optimal unimodal classifiers $f_1^*, f_2^*$, there exists constants $\epsilon_1, \epsilon_2 > 0$ such that*

$$|L(f_1) - L(f_1^*)|^2 \le \epsilon_1, |L(f_2) - L(f_2^*)|^2 \le \epsilon_2 \tag{29}$$

We now restate Theorem 2 from the main text obtaining $\underline{S}_{\mathrm{U}}$, our second lower bound on synergy linking synergy to uniqueness:

**Theorem 7.** *(Lower-bound on synergy via uniqueness, same as Theorem 2) We can relate synergy $S$ and uniqueness $U$ to modality disagreement $\alpha(f_1, f_2)$ of optimal unimodal classifiers $f_1, f_2$ as follows:*

$$\underline{S}_U = \alpha(f_1, f_2) \cdot c - \max(U_1, U_2) \le S \tag{30}$$

*for some constant $c$ depending on the label dimension $|\mathcal{Y}|$ and choice of label distance function $d$.*

Theorem 7 implies that if there is substantial disagreement between the unimodal classifiers $f_1$ and $f_2$, it must be due to the presence of unique or synergistic information. If uniqueness is small, then disagreement must be accounted for by the presence of synergy, which yields a lower bound.

*Proof.* The first part of the proof is due to an intermediate result by Sridharan & Kakade (2008), which studies how multi-view agreement can help train better multiview classifiers. We restate the key proof ideas here for completeness. The first step is to relate $I_p(X_2; Y|X_1)$ to $|L(f_1^*) - L(f^*)|^2$, the difference in errors between the Bayes' optimal unimodal classifier $f_1^*$ with the Bayes' optimal multimodal classifier $f^*$ for some appropriate loss function $L$ on the label space:

$$|L(f_1^*) - L(f^*)|^2 = |\mathbb{E}_X \mathbb{E}_{Y|X_1,X_2} \ell(f^*(x_1, x_2), y) - \mathbb{E}_X \mathbb{E}_{Y|X_1} \ell(f^*(x_1, x_2), y)|^2 \tag{31}$$

$$\le |\mathbb{E}_{Y|X_1,X_2} \ell(f^*(x_1, x_2), y) - \mathbb{E}_{Y|X_1} \ell(f^*(x_1, x_2), y)|^2 \tag{32}$$

$$\le \mathrm{KL}(p(y|x_1, x_2), p(y|x_1)) \tag{33}$$

$$\le \mathbb{E}_X \mathrm{KL}(p(y|x_1, x_2), p(y|x_1)) \tag{34}$$

$$= I_p(X_2; Y|X_1), \tag{35}$$

where we used Pinsker's inequality in (33) and Jensen's inequality in (34). Symmetrically, $|L(f_2^*) - L(f^*)|^2 \le I_p(X_1; Y|X_2)$, and via the triangle inequality through the Bayes' optimal multimodal classifier $f^*$ and the inverse Lipschitz condition we obtain

$$\mathbb{E}_{p(x_1,x_2)}[d(f_1^*, f_2^*)] \le \mathbb{E}_{p(x_1,x_2)}[d(f_1^*, f^*)] + \mathbb{E}_{p(x_1,x_2)}[d(f^*, f_2^*)] \tag{36}$$

$$\le |L(f_1^*) - L(f^*)|^2 + |L(f_2^*) - L(f^*)|^2 \tag{37}$$

$$\le I_p(X_2; Y|X_1) + I_p(X_1; Y|X_2). \tag{38}$$

Next, we relate disagreement $\alpha(f_1, f_2)$ to $I_p(X_2; Y|X_1)$ and $I_p(X_1; Y|X_2)$ via the triangle inequality through the Bayes' optimal unimodal classifiers $f_1^*$ and $f_2^*$:

$$\alpha(f_1, f_2) = \mathbb{E}_{p(x_1,x_2)}[d(f_1, f_2)] \tag{39}$$

$$\le c_d \left( \mathbb{E}_{p(x_1,x_2)}[d(f_1, f_1^*)] + \mathbb{E}_{p(x_1,x_2)}[d(f_1^*, f_2^*)] + \mathbb{E}_{p(x_1,x_2)}[d(f_2^*, f_2)] \right) \tag{40}$$

$$\le c_d \left( \epsilon_1' + I_p(X_2; Y|X_1) + I_p(X_1; Y|X_2) + \epsilon_2' \right) \tag{41}$$

$$\le 2c_d(\max(I_p(X_1; Y|X_2), I_p(X_2; Y|X_1)) + \max(\epsilon_1', \epsilon_2')) \tag{42}$$

where used classifier optimality assumption for unimodal classifiers $f_1, f_2$ in (41). Finally, we use consistency equations of PID relating $U$ and $S$ in (12)-(13): to complete the proof:

$$\alpha(f_1, f_2) \le 2c_d(\max(I_p(X_1; Y|X_2), I_p(X_2; Y|X_1)) + \max(\epsilon_1', \epsilon_2')) \tag{43}$$

$$= 2c_d(\max(U_1 + S, U_2 + S) + \max(\epsilon_1', \epsilon_2')) \tag{44}$$

$$= 2c_d(S + \max(U_1, U_2) + \max(\epsilon_1', \epsilon_2')), \tag{45}$$

In practice, setting $f_1$ and $f_2$ as neural network function approximators that can achieve the Bayes' optimal risk (Hornik et al., 1989) results in $\max(\epsilon_1', \epsilon_2') = 0$, and rearranging gives us the desired inequality. $\square$

## B.5 PROOF OF NP-HARDNESS (THEOREM 3)

Our proof is based on a reduction from the restricted timetable problem, a well-known scheduling problem closely related to constrained edge coloring in bipartite graphs. Our proof description proceeds along 4 steps.

1. Description of our problem.

2. How the minimum entropy objective can engineer "classification" problems using a technique from Kovačević et al. (2015).

3. Description of the RTT problem of Even et al. (1975), how to visualize RTT as a bipartite edge coloring problem, and a simple variant we call $Q$-RTT which RTT reduces to.

4. Polynomial reduction of $Q$-RTT to our problem.

### B.5.1 FORMAL DESCRIPTION OF OUR PROBLEM

Recall that our problem was

$$\min_{r \in \Delta_{p_{1,2,12}}} H_r(X_1, X_2, Y)$$

where $\Delta_{p_{1,2,12}} = \{r \in \Delta : r(x_1, x_2) = p(x_1, x_2), r(x_i, y) = p(x_i, y)\}$. [1] Our goal is to find the minimum-entropy distribution over $\mathcal{X}_1 \times \mathcal{X}_2 \times \mathcal{Y}$ where the pairwise marginals over $(X_1, X_2)$, $(X_1, Y)$ and $(X_2, Y)$ are specified as part of the problem. Observe that this description is symmetrical, $X_i$ and $Y$ could be swapped without loss of generality.

### B.5.2 WARM UP: USING THE MIN-ENTROPY OBJECTIVE TO MIMIC MULTICLASS CLASSIFICATION

We first note the strong similarity of our min-entropy problem to the classic *min-entropy coupling problem* in two variables. There where the goal is to find the min-entropy joint distribution over $\mathcal{X} \times \mathcal{Y}$ given fixed marginal distributions of $p(x)$ and $p(y)$. This was shown to be an NP-hard problem which has found many practical applications in recent years. An approximate solution up to $1$ bit can be found in polynomial time (and is in fact the same approximation we give to our problem). Our NP-hardness proof involves has a similar flavor as Kovačević et al. (2015), which is based on a reduction from the classic subset sum problem, exploiting the min-entropy objective to enforce discrete choices.

**Subset sum** There are $d$ items with value $c_1 \ldots c_d \geq 0$, which we assume WLOG to be normalized such that $\sum_i^d c_i = 1$. Our target sum is $0 \leq T \leq 1$. The goal is to find if some subset $\mathcal{S} \subseteq [d]$ exists such that $\sum_{i \in \mathcal{S}} c_i = T$.

**Reduction from subset sum to min-entropy coupling (Kovačević et al., 2015)** Let $\mathcal{X}$ be the $d$ items and $\mathcal{Y}$ be binary, indicating whether the item was chosen. Our joint distribution is of size $|\mathcal{X}| \times |\mathcal{Y}|$. We set the following constraints on marginals.

(i) $p(x_i) = c_i$ for all $i$, (row constraints)

(ii) $p(\text{include}) = T$, $p(\text{omit}) = 1 - T$, (column constraints)

Constraints (i) split the value of each item additively into nonnegative components to be included and not included from our chosen subset, while (ii) enforces that the items included sum to $T$. Observe that the min-entropy objective $H(X, Y) = H(Y|X) + H(X)$, which is solely dependent on $H(Y|X)$ since $H(X)$ is a constant given marginal constraints on $X$. Thus, $H(Y|X)$ is nonnegative and is only equal to $0$ *if and only if* $Y$ is deterministic given $X$, i.e., $r(x_i, \text{include}) = 0$ or $r(x_i, \text{omit}) = 0$. If our subset sum problem has a solution, then this instantiation of the min-entropy coupling problem would return a deterministic solution with $H(Y|X) = 0$, which in turn corresponds to a solution in subset sum. Conversely, if subset sum has no solution, then our min-entropy coupling problem is either infeasible OR gives solutions where $H(Y|X) > 0$ strictly, i.e., $Y|X$ is non-deterministic, which we can detect and report.

---

[1] Strictly speaking, the marginals $p(x_1, x_2)$ and $p(x_i, y)$ ought to be rational. This is not overly restrictive, since in practice these marginals often correspond to empirical distributions which would naturally be rational.

**Relationship to our problem**    Observe that our joint entropy objective may be decomposed

$$H_r(X_1, X_2, Y) = H_r(Y|X_1, X_2) + H_r(X_1, X_2).$$

Given that $p(x_1, x_2)$ is fixed under $\Delta_{p_{1,2,12}}$, our objective is equivalent to minimizing $H_r(Y|X_1, X_2)$. Similar to before, we know that $H_r(Y|X_1, X_2)$ is nonnegative and equal to zero if and only if $Y$ is deterministic given $(X_1, X_2)$.

Intuitively, we can use $\mathcal{X}_1, \mathcal{X}_2$ to represent vertices in a bipartite graph, such that $(X_1, X_2)$ are edges (which may or may not exist), and $\mathcal{Y}$ as colors for the edges. Then, the marginal constraints for $p(x_1, x_2)$ could be used alongside the min-entropy objective to ensure that each edge has exactly one color. The marginal constraints $p(x_1, y)$ and $p(x_2, y)$ tell us (roughly speaking) the number of edges of each color that is adjacent to vertices in $\mathcal{X}_1$ and $\mathcal{X}_2$.

However, this insight alone is not enough; first, edge coloring problems in bipartite graphs (e.g., colorings in regular bipartite graphs) can be solved in polynomial time, so we need a more difficult problem. Second, we need an appropriate choice of marginals for $p(x_i, y)$ that does not immediately 'reveal' the solution. Our proof uses a reduction from the *restricted timetable problem*, one of the most primitive scheduling problems available (and closely related to edge coloring or multicommodity network flow).

### B.5.3   RESTRICTED TIMETABLE PROBLEM (RTT)

The restricted timetable (RTT) problem was introduced by Even et al. (1975), and has to do with how to schedule teachers to classes they must teach. It comprises the following

- A collection of $\{T_1, \ldots, T_n\}$, where $T_i \subseteq [3]$. These represent $n$ teachers, each of which is available for the hours given in $T_i$.
- $m$ students, each of which is available at any of the 3 hours
- An binary matrix $\{0, 1\}^{n \times m}$. $R_{ij} = 1$ if teacher $i$ is required to teach class $j$, and 0 otherwise. Since $R_{ij}$ is binary, each class is taught by a teacher *at most once*.
- Each teacher is *tight*, i.e., $|T_i| = \sum_{j=1}^m R_{ij}$. That is, every teacher must teach whenever they are available.

Suppose there are exactly 3 hours a day. The problem is to determine if there exists a meeting function

$$f : [n] \times [m] \times [3] \to \{0, 1\},$$

where our goal is to have $f(i, j, h) = 1$ if and only if teacher $i$ teaches class $j$ at the $h$-th hour. We require the following conditions in our meeting function:

1. $f(i, j, h) = 1 \implies h \in T_i$. This implies that teachers are only teaching in the hours they are available.
2. $\sum_{h \in [3]} f(i, j, h) = R_{ij}$ for all $i \in [n], j \in [m]$. This ensures that every class gets the teaching they are required, as specified by $R$.
3. $\sum_{i \in [n]} f(i, j, h) \le 1$ for all $j \in [m]$ and $h \in [3]$. This ensures no class is taught by more than one teachers at once.
4. $\sum_{j \in [m]} f(i, j, h) \le 1$ for all $i \in [n]$ and $h \in [3]$. This ensures no teacher is teaching more than one class simultaneously.

Even et al. (1975) showed that RTT is NP-hard via a clever reduction from 3-SAT. Our strategy is to reduce RTT to our problem.

**Viewing RTT through the lens of bipartite edge coloring**    RTT can be visualized as a variant of constrained edge coloring in bipartite graphs (Figure 4). The teachers and classes are the two different sets of vertices, while $R$ gives the adjacency structure. There are 3 colors available, corresponding to hours in a day. The task is to color the edges of the graph with these 3 colors such that

1. No two edges of the same color are adjacent. This ensures students and classes are at most teaching/taking one session at any given hour (condition 3 and 4)
2. Edges adjacent to teacher $i$ are only allowed colors in $T_i$. This ensures teachers are only teaching in available hours (condition 1)

If every edge is colored while obeying the above conditions, then it follows from the tightness of teachers (in the definition of RTT) that every class is assigned their required lessons (condition 2). The decision version of the problem is to return if such a coloring is possible.

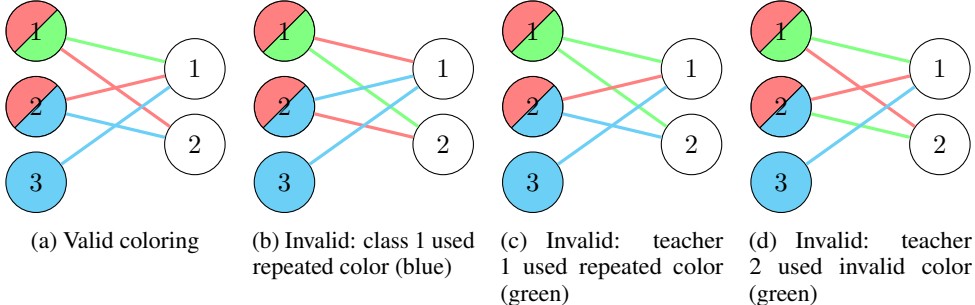

(a) Valid coloring | (b) Invalid: class 1 used repeated color (blue) | (c) Invalid: teacher 1 used repeated color (green) | (d) Invalid: teacher 2 used invalid color (green)

Figure 4: Examples of valid and invalid colorings. Left vertices are teachers 1, 2, 3. Right vertices are classes 1, 2. The colors red, green, blue are for hours 1, 2, 3 respectively, color of teacher vertices are the hours where the teachers are available (by definition of RTT, the number of distinct colors per teacher vertex is equal to its degree). The color of an edge (red, green or blue) says that a teacher is assigned to that class at that hour. Figure 4a shows a valid coloring (or timetabling), since (i) all edges are colored, (ii) no edge of the same colors are adjacent, and (iii) edges adjacent to teachers correspond to the vertex's color. Figures 4b, 4c, 4d are invalid colorings because of same-colored edges being adjacent, or teacher vertex colors differing to adjacent edges.

**Time Constrained RTT ($Q$-RTT)**  A variant of RTT that will be useful is when we impose restrictions on the number of classes being taught at any each hour. We call this $Q$-RTT, where $Q = (q_1, q_2, q_3) \in \mathbb{Z}^3$. $Q$-RTT returns true if, in addition to the usual RTT conditions, we require the meeting function to satisfy

$$\sum_{i \in [n], j \in [m]} f(i, j, h) = q_h.$$

That is, the total number of hours taught by teachers in hour $h$ is exactly $q_h$. From the perspective of edge coloring, $Q$-RTT simply imposes an additional restriction on the total number of edges of each color, i.e., there are $q_k$ edges of color $k$ for each $k \in [3]$.

Obviously, RTT can be Cook reduced to $Q$-RTT: since there are only 3 hours and a total of $g = \sum_{i \in [n], j \in [m]} R_{ij}$ total lessons to be taught, there are at most $\mathcal{O}(g^2)$ ways of splitting the required number of lessons up amongst the 3 hours. Thus, we can solve RTT by making at most $\mathcal{O}(g^2)$ calls to $Q$-RTT. This is polynomial in the size of RTT, and we conclude $Q$-RTT is NP-hard.

### B.5.4 REDUCTION OF $Q$-RTT TO OUR PROBLEM

We will reduce $Q$-RTT to our problem. Let $\alpha = 1/\left(\sum_{i,j} R_{ij} + 3m\right)$, where $1/\alpha$ should be seen as a normalizing constant given by the number of edges in a bipartite graph. One should think of $\alpha$ as an indicator of the boolean TRUE and $0$ as FALSE. We use the following construction

1. Let $\mathcal{X}_1 = [n] \cup \mathcal{Z}$, where $\mathcal{Z} = \{Z_1, Z_2, Z_3\}$. From a bipartite graph interpretation, these form one set of vertices that we will match to classes. $Z_1, Z_2, Z_3$ are "holding rooms", one for each of the 3 hours. Holding rooms are like teachers whose classes can be assigned in order to pass the time. They will not fulfill any constraints on $R$, but they *can* accommodate multiple classes at once. We will explain the importance of these holding rooms later.

2. Let $\mathcal{X}_2 = [m]$. These form the other set of vertices, one for each class.

3. Let $\mathcal{Y} = [3] \cup \{0\}$. 1, 2, and 3 are the 3 distinct hours, corresponding to edge colors. 0 is a special "null" color which will only be used when coloring edges adjacent to the holding rooms.

4. Let $p(i, j, \cdot) = \alpha \cdot R_{ij}$ and $p(i, j) = \alpha$ for all $i \in \mathcal{Z}$, $j \in [m]$. Essentially, there is an edge between a teacher and class if $R$ dictates it. There are also *always* edges from every holding room to each class.

5. For $i \in [n]$, set $p(i, \cdot, h) = \alpha$ if $h \in T_i$, 0 otherwise. For $Z_i \in \mathcal{Z}$, we set

$$p(Z_i, \cdot, h) = \begin{cases} \alpha \cdot q_i & h = 0 \\ \alpha \cdot (m - q_i) & h = i \\ 0 & \text{otherwise} \end{cases}$$

In order words, at hour $h$, when a class is not assigned to some teacher (which would to contribute to $q_h$), they must be placed in holding room $Z_h$.

6. Let $p(\cdot, j, h) = \alpha$ for $h \in [3]$, and $p(\cdot, j, h) = \alpha \cdot \sum_{i \in [n]} R_{i,j}$. The former constraint means that for each of the 3 hours, the class must be taking some lesson with a teacher OR in the holding room. The second constraint assigns the special "null" value to the holding rooms which were not used by that class.

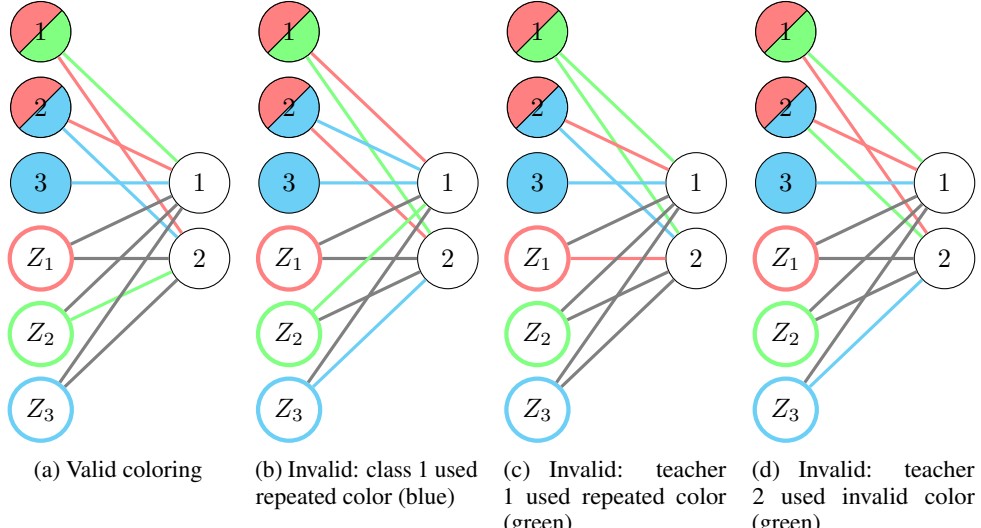

(a) Valid coloring

(b) Invalid: class 1 used repeated color (blue)

(c) Invalid: teacher 1 used repeated color (green)

(d) Invalid: teacher 2 used invalid color (green)

Figure 5: Examples of valid and invalid colorings when holding rooms are included. For simplicity, we illustrate all constraints except those on $Q$. Left vertices are teachers 1, 2, 3 and holding rooms $Z_1, Z_2, Z_3$. Right vertices are classes 1, 2. The colors red, green, blue are for hours 1, 2, 3 respectively, color of teacher vertices are the hours where the teachers are available (by definition of RTT, the number of distinct colors per teacher vertex is equal to its degree). Border color of holding room vertices are the hour that the holding room is available. The color of an edge (red, green or blue) says that a teacher (or holding room) is assigned to that class at that hour. Gray edges are the "null" color, meaning that that waiting room is not used by that class. Figure 5a shows a valid coloring (or timetabling), since all edges are colored, no edge of the same colors are adjacent (other than the gray ones), and edges adjacent to teachers correspond to the vertex's color. Figures 5b, 5c, 5d are invalid colorings because of non-gray edges being adjacent, or teacher vertices being adjacent to colors different from itself.

**A solution to our construction with 0 conditional entropy implies a valid solution to $Q$-RTT**
Suppose that our construction returns a distribution $r$ such that every entry $r(x_1, x_2, y)$ is either $\alpha$ or 0. We claim that the meeting function $f(i, j, h) = 1$ if $r(i, j, h) = \alpha$ and 0 otherwise solves $Q$-RTT.

- Teachers are only teaching in the hours they are available, because of our marginal constraint on $p(i, \cdot, h)$.

- Every class gets the teaching they need. This follows from the fact that teachers are tight and the marginal constraint $p(i, \cdot, h)$, which forces teachers to be teaching whenever they can. The students are getting the lessons from the right teachers because of the marginal constraint on $p(i, j, \cdot)$, since teachers who are not supposed to teach a class have those marginal values set to 0.

- No class is taught by more than one teacher at once. This follows from marginal constraint $p(\cdot, j, h)$. For each of the hours, a class is with either a single teacher or the holding room.

- No teacher is teaching more than one class simultaneously. This holds again from our marginal constraint on $p(i, \cdot, h)$.

- Lastly, the total number of lessons (not in holding rooms) held in each hour is $q_h$ as required by $Q$-RTT. To see why, we consider each color (hour). Each color (excluding the null color) is used exactly $m$ times by virtue of $p(\cdot, j, h)$. Some of these are in holding rooms, other are

with teachers. The former (over all classes) is given by $m - q_h$ because of our constraint on $p(i, \cdot, h)$, which means that exactly $q_h$ lessons in hour $h$ as required.

**A valid solution to $Q$-RTT implies a solution to our construction with 0 conditional entropy**
Given a solution to $Q$-RTT, we recover a candidate solution to our construction in a natural way. If teacher $i$ is teaching class $j$ in hour $h$, then color edge $ij$ with color $h$, i.e., $r(i, j, h) = \alpha$ and $r(i, j, h') = 0$ if $h' \neq h$. Since in RTT each teacher and class can be assigned one lesson per hour at most, there will be no clashes with this assignment. For all other $i \in [3], j \in [m]$ where $R_{ij} = 0$, we assign $r(i, j, \cdot) = 0$. Now, we will also need to assign students to holding rooms. For $h \in [3]$, we set $r(Z_h, j, h) = \alpha$ if class $j$ was not assigned to any teacher in hour $h$. If class $j$ was assigned some teacher in hour $h$, then $r(Z_h, j, 0) = \alpha$, i.e., we give it the special null color. All other entries are given a value of $0$. We can verify

- $r$ is a valid probability distribution. The nonnegativity of $r$ follows from the fact that $\alpha > 0$ strictly. We need to check that $r$ sums to $1$. We break this down into two cases based on whether the first argument of $r$ is some $Z_h$ or $i$. In Case 1, we have

$$\sum_{i \in [n], h \in [3] \cup \{0\}, j \in [m]} r(i, j, h) = \sum_{i \in [n], h \in [3], j \in [m]} r(i, j, h)$$
$$= \alpha \cdot \sum_{i \in [n], j \in [m]} R_{ij},$$

where the first line follows from the fact that we never color a teacher-class edge with the null color, and the second line is because every class gets its teaching requirements satisfied. In Case 2, we know that by definition every class is matched to every holding room and assigned either the null color or that room's color, hence

$$\sum_{i \in \{Z_1, Z_2, Z_3\}, h \in [3] \cup \{0\}, j \in [m]} r(i, j, h) = 3m$$

Summing them up, we have $\alpha \cdot \left(3m + \sum_{i \in [n], j \in [m]} R_{ij}\right) = 1$ (by our definition of $\alpha$.

- This $r$ distribution has only entries in $\alpha$ or $0$. This follows by definition.

- This $r$ distribution has minimum conditional entropy. For a fixed $i, j$, $r(i, j, \cdot)$ is either $\alpha$ or $0$. That is, $Y$ is deterministic given $X_1, X_2$, hence $H(Y|X_1, X_2) = 0$.

- All 3 marginal constraints in our construction are obeyed. We check them in turn.

  - Marginal constraint $r(i, j) = p(i, j)$. When $i \in [3]$: (i) when $R_{ij} = 1$ exactly one time $h$ is assigned to teacher $i$ and class $j$, hence $r(i, j) = \alpha = p(i, j)$ as required, (ii)when $R_{ij} = 0$ as specified. Now when $i \in \{Z_1, Z_2, Z_3\}$, we have $r(i, j, \cdot) = \alpha = p(i, j)$ since every holding room is either assigned it's color to a class, or assigned the special null color.

  - Marginal constraint $r(i, h) = p(i, h)$. When $i \in [3]$, this follows directly from tightness. Similarly, when $i \in \{Z_1, Z_2, Z_3\}$, we have by definition of $Q$-RTT the assignments to holding rooms equal to $m - q_h$ for hour $h$, and consequently, $q_h$ null colors adjacent to $Z_h$ as required.

  - Marginal constraint $r(j, h) = p(j, h)$. For every $h \in [3]$, the class is assigned either to a teacher or a holding room, so this is equal to $\alpha$ as required. For $h = 0$, i.e., the null color, this is used exactly $\sum_{i \in [n]} R_{ij}$ times (since these were the number hours that were *not* assigned to teachers), as required, making its marginal $\sum_{i \in [n]} R_{ij}$ and $r(j, h) = \alpha \cdot \sum_{i \in [n]} R_{ij}$ as required.

Thus, if RTT returns TRUE, our construction will also return a solution with entries in $\{0, \alpha\}$, and vice versa.

**Corollary** The decision problem of whether there exists a distribution in $r \in \Delta_{p_{1,2,12}}$ such that $H(Y|X_1, X_2) = 0$ is NP-complete. This follows because the problem is in NP since checking if $Y$ is deterministic (i.e., $H(Y|X_1, X_2) = 0$) can be done in polynomial time, while NP-hardness follows from the same argument as above.

### B.6 Upper bound on synergy (Theorem 4)

We begin by restating Theorem 4 from the main text:

**Theorem 8.** *(Upper-bound on synergy, same as Theorem 4).*

$$S \leq H_p(X_1, X_2) + H_p(Y) - \min_{r \in \Delta_{p_{12,y}}} H_r(X_1, X_2, Y) - \max_{q \in \Delta_{p_{1,2}}} I_q(\{X_1, X_2\}; Y) = \overline{S} \qquad (46)$$

where $\Delta_{p_{12,y}} = \{r \in \Delta : r(x_1, x_2) = p(x_1, x_2), r(y) = p(y)\}$.

*Proof.* Recall that this upper bound boils down to finding $\max_{r \in \Delta_{p_{1,2,12}}} I_r(\{X_1, X_2\}; Y)$. We have

$$\max_{r \in \Delta_{p_{1,2,12}}} I_r(\{X_1, X_2\}; Y) = \max_{r \in \Delta_{p_{1,2,12}}} \{H_r(X_1, X_2) + H_r(Y) - H_r(X_1, X_2, Y)\} \qquad (47)$$

$$= H_p(X_1, X_2) + H_p(Y) - \min_{r \in \Delta_{p_{1,2,12}}} H_r(X_1, X_2, Y), \qquad (48)$$

$$\leq H_p(X_1, X_2) + H_p(Y) - \min_{r \in \Delta_{p_{12,y}}} H_r(X_1, X_2, Y) \qquad (49)$$

where the first two lines are by definition. The last line follows since $\Delta_{p_{12,y}}$ is a superset of $r \in \Delta_{p_{1,2,12}}$, which implies that minimizing over it would yield a a no larger objective. $\square$

In practice, we use the slightly tighter bound which maximizes over all the pairwise marginals,

$$\max_{r \in \Delta_{p_{1,2,12}}} I_r(X_1, X_2; Y) \leq H_p(X_1, X_2) + H_p(Y) - \max \begin{cases} \min_{r \in \Delta_{p_{12,y}}} H_r(X_1, X_2, Y) \\ \min_{r \in \Delta_{p_{1,x_2}}} H_r(X_1, X_2, Y) \\ \min_{r \in \Delta_{p_{2,x_1}}} H_r(X_1, X_2, Y) \end{cases}. \qquad (50)$$

**Estimating $\overline{S}$ using min-entropy couplings** We only show how to compute $\min_{r \in \Delta_{p_{12,y}}} H_r(X_1, X_2, Y)$, since the other variants can be computed in the same manner via symmetry. We recognize that by treating $(X_1, X_2) = X$ as a single variable, we recover the classic *min-entropy coupling* over $X$ and $Y$, which is still NP-hard but admits good approximations (Cicalese & Vaccaro, 2002; Cicalese et al., 2017; Kocaoglu et al., 2017; Rossi, 2019; Compton, 2022; Compton et al., 2023).

There are many methods to estimate such a coupling, for example Kocaoglu et al. (2017) give a greedy algorithm running in linear-logarithmic time, which was further proven by Rossi (2019); Compton (2022) to be a 1-bit approximation of the minimum coupling [2]. Another line of work was by (Cicalese et al., 2017), which constructs an appropriate coupling and shows that it is optimal to 1-bit to a lower bound $H(p(x_1, x_2) \wedge p(y))$, where $\wedge$ is the greatest-lower-bound operator, which they showed in (Cicalese & Vaccaro, 2002) can be computed in linear-logarithmic time. We very briefly describe this method; more details may be found in (Cicalese et al., 2017; Cicalese & Vaccaro, 2002) directly.

**Remark** A very recent paper by Compton et al. (2023) show that one can get an approximation tighter than 1-bit. We leave the incorporation of these more advanced methods as future work.

Without loss of generality, suppose that $\mathcal{X}$ and $\mathcal{Y}$ are ordered and indexed such that $p(x)$ and $p(y)$ are sorted in non-increasing order of the marginal constraints, i.e., $p(X = x_i) \geq p(X = x_j)$ for all $i \leq j$. We also assume WLOG that the supports of $X$ and $Y$ are of the same size $n$, if they are not, then pad the smaller one with dummy values and introduce marginals that constrain these values to never occur (and set $n$ accordingly if needed). For simplicity, we will just refer to $p_i$ and $q_j$ for the distributions of $p(X = x_i)$ and $p(Y = y_j)$ respectively.

Given 2 distributions $p, q$ we say that $p$ is majorized by $q$, written as $p \preceq q$ if and only if

$$\sum_{i=1}^{k} p_i \leq \sum_{i=1}^{k} q_i \qquad \text{for all } k \in 1 \ldots n \qquad (51)$$

---

[2]This a special case when there are 2 modalities. For more modalities, the bounds will depend on the sizes and number of signals.

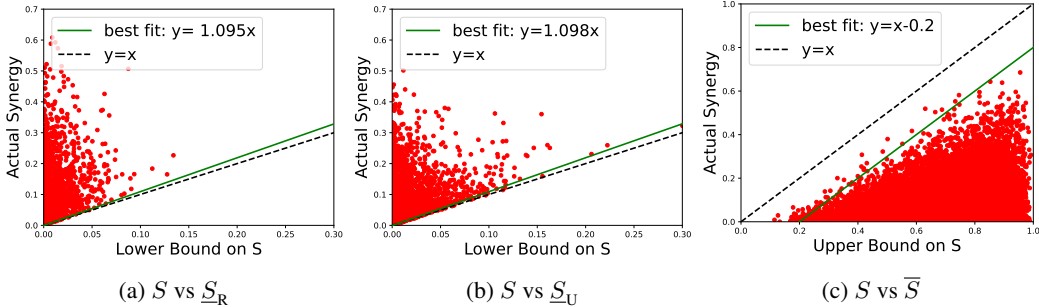

(a) $S$ vs $\underline{S}_\text{R}$        (b) $S$ vs $\underline{S}_\text{U}$        (c) $S$ vs $\overline{S}$

Figure 6: Plots of actual synergy against our estimated (a) lower bound on synergy $\underline{S}_\text{R}$, (b) lower bound on synergy $\underline{S}_\text{R}$, and (c) upper bound $\overline{S}$. The bounds closely track true synergy, which we show via three lines of best fit that almost exactly track true synergy: $y = 1.095x$, $y = 1.098x$, and $y = x - 0.2$.

As Cicalese & Vaccaro (2002) point out, there is a strong link between majorization and Schur-convex functions; in particular, if $p \preceq q$, then we have $H(p) \geq H(q)$. Indeed, if we treat $\succeq$ as a partial order and consider the set

$$\mathcal{P}^n = \left\{ p = (p_1, \ldots, p_n) : p_i \in [0,1], \sum_i^n p_i = 1, p_i \geq p_{i+1} \right\}$$

as the set of finite (ordered) distributions with support size $n$ with non-increasing probabilities, then we obtain a lattice with a unique greatest lower bound ($\wedge$) and least upper bound ($\vee$). Then, Cicalese & Vaccaro (2002) show that that $p \wedge q$ can be computed recursively as $p \wedge q = \alpha(p, q) = (a_1, \ldots, a_n)$ where

$$a_i = \min\{ \sum_{j=1}^i p_j, \sum_{j=1}^i q_j \} + \sum_{j=1}^{i-1} a_{j-1}$$

It was shown by Cicalese et al. (2017) that *any* coupling satisfying the marginal constraints given by $p$ and $q$, i.e.,

$$M \in C(p, q) = \left\{ M = m_{ij} : \sum_j m_{ij} = p_i, \sum_i m_{ij} = p_j \right\}$$

has entropy $H(M) \geq H(p \wedge q)$. In particular, this includes the min-entropy one. Since we only need the optimal value of such a coupling and not the actual coupling per-se, we can use plug the value of $H(p \wedge q)$ into the minimization term (49), which yields an upper bound for $\max_{r \in \Delta_{p_{1,2,12}}} I_r(\{X_1, X_2\}; Y)$, which would form an upper bound on $\overline{S}$ itself.

## C    EXPERIMENTAL DETAILS

### C.1    VERIFYING LOWER AND UPPER BOUNDS

**Synthetically generated datasets**: To test our derived bounds on synthetic data, We randomly sampled $100,000$ distributions of $\{X_1, X_2, Y\}$ to calculate their bounds and compare with their actual synergy values. We set $X_1, X_2$, and $Y$ as random binary values, so each distribution can be represented as a size 8 vector of randomly sampled entries that sum up to 1.

**Results**: We calculated the lower bound via redundancy, lower bound via disagreement, and upper bound of all distributions and plotted them with actual synergy value (Figure 6). We define a distribution to be on the boundary if its lower/upper bound is within $10\%$ difference from its actual synergy value. We conducted the least mean-square-error fitting on these distributions close to the boundary. We plot actual synergy against $\underline{S}_\text{R}$ in Figure 6 (left), and find that it again tracks a lower bound of synergy. In fact, we can do better and fit a linear trend $y = 1.095x$ on the distributions along the margin (RMSE = 0.0013).

We also plot actual synergy against computed $\underline{S}_\text{U}$ in Figure 6 (middle). As expected, the lower bound closely tracks actual synergy. Similarly, we can again fit a linear model on the points along the boundary, obtaining $y = 1.098x$ with a RMSE of $0.0075$ (see this line in Figure 6 (middle)).

Finally, we plot actual synergy against estimated $\overline{S}$ in Figure 6 (right). Again, we find that the upper bound consistently tracks the highest attainable synergy - we can fit a single constant $y = x - 0.2$ to

obtain an RMSE of 0.0022 (see this line in Figure 6 (right)). This implies that our bound enables both accurate comparative analysis of relative synergy across different datasets, and precise estimation of absolute synergy.

**Real-world datasets**: We also use the large collection of real-world datasets in MultiBench (Liang et al., 2021): (1) MOSI: video-based sentiment analysis (Zadeh et al., 2016), (2) MOSEI: video-based sentiment and emotion analysis (Zadeh et al., 2018), (3) MUSTARD: video-based sarcasm detection (Castro et al., 2019), (5) MIMIC: mortality and disease prediction from tabular patient data and medical sensors (Johnson et al., 2016), and (6) ENRICO: classification of mobile user interfaces and screenshots (Leiva et al., 2020). While the previous bitwise datasets with small and discrete support yield exact lower and upper bounds, this new setting with high-dimensional continuous modalities requires the approximation of disagreement and information-theoretic quantities: we train unimodal neural network classifiers $\hat{f}_\theta(y|x_1)$ and $\hat{f}_\theta(y|x_2)$ to estimate disagreement, and we cluster representations of $X_i$ to approximate the continuous modalities by discrete distributions with finite support to compute lower and upper bounds.

**Implementation details**: We first apply PCA to reduce the dimension of multimodal data. For the test split, we use unsupervised clustering to generate 20 clusters. We obtain a clustered version of the original dataset $\mathcal{D} = \{(x_1, x_2, y)\}$ as $\mathcal{D}_{\text{cluster}} = \{(c_1, c_2, y)\}$ where $c_i \in \{1, \dots, 20\}$ is the ID of the cluster that $x_i$ belongs to. In our experiments, where $\mathcal{Y}$ is typically a classification task, we set the unimodal classifiers $f_1 = \hat{p}(y|x_1)$ and $f_2 = \hat{p}(y|x_2)$ as the Bayes optimal classifiers for multiclass classification tasks.

For classification, $\mathcal{Y}$ is the set of $k$-dimensional 1-hot vectors. Given two logits $\hat{y}_1, \hat{y}_2$ obtained from $x_1, x_2$ respectively, define $d(\hat{y}_1, \hat{y}_2) = (\hat{y}_1 - \hat{y}_2)^2$. We have that $c_d = 1$, and $\epsilon_1 = |L(f_1) - L(f_1^*)|^2 = 0$ and $\epsilon_2 = |L(f_2) - L(f_2^*)|^2 = 0$ for well-trained neural network unimodal classifiers $f_1$ and $f_2$ for Theorem 2. For datasets with 3 modalities, we perform the experiments separately for each of the 3 modality pairs, before taking an average over the 3 modality pairs. Extending the definitions of redundancy, uniqueness, and synergy, as well as our derived bounds on synergy for 3 or more modalities is an important open question for future work.

## C.2 RELATIONSHIPS BETWEEN AGREEMENT, DISAGREEMENT, AND INTERACTIONS

**1. The relationship between synergy and redundancy**: We give some example distributions to analyze when the lower bound based on redundancy $\underline{S}_{\text{R}}$ is high or low. The bound is high for distributions where $X_1$ and $X_2$ are independent, but $Y = 1$ sets $X_1 \neq X_2$ to increase their dependence (i.e., AGREEMENT XOR distribution in Table 2b). Since $X_1$ and $X_2$ are independent but become dependent given $Y$, $I(X_1; X_2; Y)$ is negative, and the bound is tight $\underline{S}_{\text{R}} = 1 \leq 1 = S$. Visual Question Answering 2.0 (Goyal et al., 2017) falls under this category, with $S = 4.92, R = 0.79$, where the image and question are independent (some questions like 'what is the color of the object' or 'how many people are there' can be asked for many images), but the answer connects the two modalities, resulting in dependence given the label. As expected, the estimated lower bound for synergy: $\underline{S}_{\text{R}} = 4.03 \leq 4.92 = S$.

Conversely, the bound is low for Table 2d with the probability mass distributed uniformly only when $y = x_1 = x_2$ and 0 elsewhere. As a result, $X_1$ is always equal to $X_2$ (perfect dependence), and yet $Y$ perfectly explains away the dependence between $X_1$ and $X_2$ so $I(X_1; X_2|Y) = 0$: $\underline{S}_{\text{R}} = 0 \leq 0 = S$. Note that this is an example of perfect redundancy and zero synergy - for an example with synergy, refer back to DISAGREEMENT XOR in Table 2a - due to disagreement there is non-zero $I(X_1; X_2)$ but the label explains some of the relationships between $X_1$ and $X_2$ so $I(X_1; X_2|Y) < I(X_1; X_2)$: $\underline{S}_{\text{R}} = -0.3 \leq 1 = S$. A real-world example is multimodal sentiment analysis from text, video, and audio of monologue videos on MOSEI, $R = 0.26$ and $S = 0.04$, and as expected the lower bound is small $\underline{S}_{\text{R}} = 0.01 \leq 0.04 = S$.

**2. The relationship between synergy and uniqueness**: To give an intuition of the relationship between disagreement, uniqueness, and synergy, we use one illustrative example shown in Table 2a, which we call DISAGREEMENT XOR. We observe that there is maximum disagreement between marginals $p(y|x_1)$ and $p(y|x_2)$: the likelihood for $y$ is high when $y$ is the same bit as $x_1$, but reversed for $x_2$. Given both $x_1$ and $x_2$: $y$ seems to take a 'disagreement' XOR of the individual marginals, i.e. $p(y|x_1, x_2) = p(y|x_1)$ XOR $p(y|x_2)$, which indicates synergy (note that an exact XOR would imply perfect agreement and high synergy). The actual disagreement is 0.15, synergy is 0.16, and uniqueness is 0.02, indicating a very strong lower bound $\underline{S}_{\text{U}} = 0.13 \leq 0.16 = S$. A real-world

Table 4: We show the full list of computed lower and upper bounds on $S$ without labeled multimodal data and compare them to the true $S$ assuming knowledge of the full joint distribution $p$: the bounds track $S$ well on MUSTARD and MIMIC, and also show general trends on the other datasets except ENRICO where estimating synergy is difficult. V = video, T = text, A = audio modalities.

| | MOSEI$_{V+T}$ | MOSEI$_{V+A}$ | MOSEI$_{A+T}$ | UR-FUNNY$_{V+T}$ | UR-FUNNY$_{V+A}$ | UR-FUNNY$_{A+T}$ |
|---|---|---|---|---|---|---|
| $\overline{S}$ | 0.96 | 0.98 | 0.97 | 0.96 | 0.96 | 0.99 |
| $S$ | 0.04 | 0.03 | 0.03 | 0.21 | 0.24 | 0.08 |
| $\underline{S}_{\mathrm{R}}$ | 0.01 | 0.0 | 0.0 | 0.0 | 0.0 | 0.0 |
| $\underline{S}_{\mathrm{U}}$ | 0.01 | 0.01 | 0.0 | 0.0 | 0.0 | 0.01 |

| | MOSI$_{V,T}$ | MOSI$_{V,A}$ | MOSI$_{A,T}$ | MUSTARD$_{V,T}$ | MUSTARD$_{V,A}$ | MUSTARD$_{A,T}$ | MIMIC | ENRICO |
|---|---|---|---|---|---|---|---|---|
| $\overline{S}$ | 0.92 | 0.92 | 0.93 | 0.79 | 0.78 | 0.79 | 0.41 | 2.09 |
| $S$ | 0.31 | 0.28 | 0.14 | 0.49 | 0.31 | 0.51 | 0.02 | 1.02 |
| $\underline{S}_{\mathrm{R}}$ | 0.01 | 0.01 | 0.0 | 0.04 | 0.01 | 0.06 | 0.0 | 0.01 |
| $\underline{S}_{\mathrm{U}}$ | 0.03 | 0.03 | 0.02 | 0.07 | 0.06 | 0.11 | −0.12 | −0.55 |

equivalent dataset is MUSTARD for sarcasm detection from video, audio, and text (Castro et al., 2019), where the presence of sarcasm is often due to a contradiction between what is expressed in language and speech, so disagreement $\alpha = 0.12$ is the highest out of all the video datasets, giving a lower bound $\underline{S}_{\mathrm{U}} = 0.11 \leq 0.44 = S$.

On the contrary, the lower bound is low when all disagreement is explained by uniqueness (e.g., $y = x_1$, Table 2c), which results in $\underline{S}_{\mathrm{U}} = 0 \leq 0 = S$ ($\alpha$ and $U$ cancel each other out). A real-world equivalent is MIMIC involving mortality and disease prediction from tabular patient data and time-series medical sensors (Johnson et al., 2016). Disagreement is high $\alpha = 0.13$ due to unique information $U_1 = 0.25$, so the lower bound informs us about the lack of synergy $\underline{S}_{\mathrm{U}} = -0.12 \leq 0.02 = S$.

Finally, the lower bound is loose when there is synergy without disagreement, such as AGREEMENT XOR ($y = x_1$ XOR $x_2$, Table 2b) where the marginals $p(y|x_i)$ are both uniform, but there is full synergy: $\underline{S}_{\mathrm{U}} = 0 \leq 1 = S$. Real-world datasets that have synergy without disagreement include UR-FUNNY where there is low disagreement in predicting humor $\alpha = 0.03$, and relatively high synergy $S = 0.18$, which results in a loose lower bound $\underline{S}_{\mathrm{U}} = 0.01 \leq 0.18 = S$.

**3. On upper bounds for synergy**: We also run experiments to obtain estimated upper bounds on synthetic and MultiBench datasets. The quality of the upper bound shows some intriguing relationships with that of lower bounds. For distributions with perfect agreement and synergy such as $y = x_1$ XOR $x_2$ (Table 2b), $\overline{S} = 1 \geq 1 = S$ is really close to true synergy, $\underline{S}_{\mathrm{R}} = 1 \leq 1 = S$ is also tight, but $\underline{S}_{\mathrm{U}} = 0 \leq 1 = S$ is loose. For distributions with disagreement and synergy (Table 2a), $\overline{S} = 0.52 \geq 0.13 = S$ far exceeds actual synergy, $\underline{S}_{\mathrm{R}} = -0.3 \leq 1 = S$ is much lower than actual synergy, but $\underline{S}_{\mathrm{U}} = 0.13 \leq 0.16 = S$ is tight (see relationships in Figure 7).

Finally, while some upper bounds (e.g., MUSTARD, MIMIC) are close to true $S$, some of the other examples in Table 1 show bounds that are quite weak. This could be because (i) there indeed exists high synergy distributions that match $\mathcal{D}_i$ and $\mathcal{D}_M$, but these are rare in the real world, or (ii) our approximation used in Theorem 4 is mathematically loose. We leave these for future work.

Figure 7: Comparing the qualities of the bounds when there is agreement, disagreement, and synergy. When there is agreement and synergy, $\underline{S}_{\mathrm{R}}$ is tight, $\underline{S}_{\mathrm{U}}$ is loose, and $\overline{S}$ is tight. When there is disagreement and synergy, $\underline{S}_{\mathrm{R}}$ is loose, $\underline{S}_{\mathrm{U}}$ is tight, and $\overline{S}$ is loose with respect to true $S$.

## C.3    COMPARISONS WITH OTHER INTERACTION MEASURES

We are not aware of other related work in mathematically formalizing multimodal interactions, much less for semi-supervised data. Most of our results are new theoretical and empirical insights about different multimodal interactions. Other valid definitions of multimodal interactions do exist, but are known to suffer from issues regarding over- and under-estimation, and may even be negative (Griffith & Koch, 2014).

In Table 5, we compare our estimators with other previously used measures for feature interactions on the synthetic datasets. We implement 6 other definitions: (1) **I-min** can sometimes overestimate

Table 5: Estimating multimodal interactions on synthetic generative model datasets. Ground truth total information is computed based on an upper bound from the best multimodal test accuracy. Our estimated interactions are consistent with ground truth interactions. We also implement 3 other definitions: **I-min** can sometimes overestimate synergy and uniqueness; **WMS** is actually synergy minus redundancy, so can be negative and when $R$ & $S$ are of equal magnitude WMS cancels out to be 0; **CI** can also be negative and sometimes incorrectly concludes highest uniqueness for S-only data.

| Task | $R$-only data | | | | $U_1$-only data | | | | $U_2$-only data | | | | $S$-only data | | | |
|---|---|---|---|---|---|---|---|---|---|---|---|---|---|---|---|---|
| Interaction | $R$ | $U_1$ | $U_2$ | $S$ | $R$ | $U_1$ | $U_2$ | $S$ | $R$ | $U_1$ | $U_2$ | $S$ | $R$ | $U_1$ | $U_2$ | $S$ |
| I-MIN | 0.17 | 0.08 | 0.07 | 0 | 0 | 0.23 | 0 | 0.06 | 0 | 0 | 0.25 | 0.08 | 0.07 | 0.03 | 0.04 | **0.17** |
| WMS | 0 | 0.20 | 0.20 | −0.11 | 0 | 0.25 | 0.02 | 0.05 | 0 | 0.03 | **0.27** | 0.05 | 0 | 0.14 | 0.15 | 0.07 |
| CI | **0.34** | −0.09 | −0.10 | 0.17 | 0 | 0.23 | 0 | 0.06 | 0 | 0.01 | 0.25 | 0.07 | −0.02 | 0.13 | 0.14 | 0.08 |
| **Ours** | 0.16 | 0 | 0 | 0.05 | 0 | 0.16 | 0 | 0.05 | 0 | 0 | 0.17 | 0.05 | 0.07 | 0 | 0.01 | **0.14** |
| Truth | 0.58 | 0 | 0 | 0 | 0 | 0.56 | 0 | 0 | 0 | 0 | 0.54 | 0 | 0 | 0 | 0 | 0.56 |

synergy and uniqueness, (2) **WMS** is actually synergy minus redundancy, so can be negative and when $R$ & $S$ are of equal magnitude WMS cancels out to be 0, (3) **CI** can also be negative and sometimes incorrectly concludes highest uniqueness for $S$-only data, (4) **Shapley values**, (5) **Integrated gradients (IG)**, and (6) **CCA**, which are based on quantifying interactions captured by a multimodal model. Our work is fundamentally different in that interactions are properties of data before training any models. Our estimated interactions are more consistent with ground truth interactions.

### C.4 Robustness to imperfect unimodal classifiers

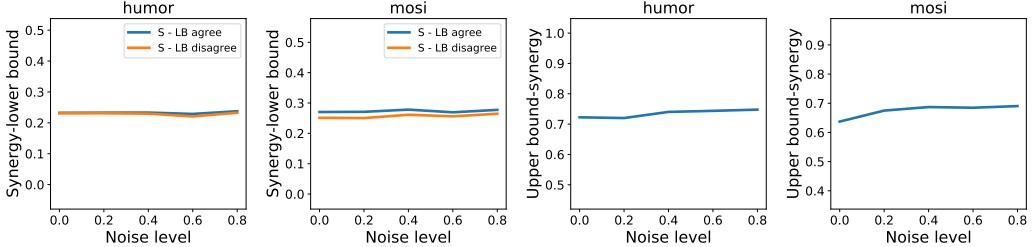

Figure 8: We study the effect of noisy unimodal predictors and disagreement by perturbing the label by various noise levels (x-axis from noise = 0.0 to 0.8) and examining the change in estimated upper and lower bounds (left: y-axis is true synergy - lower bound and right: y-axis is upper bound - true synergy). On 2 real-world datasets (UR-FUNNY and MOSI) we find bounds quite robust to label noise, giving stable trends of real synergy.

We also study the effect of imperfect unimodal predictors and therefore imperfect disagreement measurements on our derived bounds, by perturbing the label by various noise levels (from no noise to very noisy) and examining the change in both the estimated upper and lower bounds. In Figure 8, we find these bounds to be quite robust to imperfect unimodal classifiers, still giving close trends of real synergy.

## D ESTIMATING MULTIMODAL PERFORMANCE

Formally, we estimate performance via a combination of Feder & Merhav (1994) and Fano's inequality (Fano, 1968) together yield tight bounds of performance as a function of total information $I_p(\{X_1, X_2\}; Y)$. We restate Theorem 5 from the main text:

**Theorem 9.** *Let $P_{acc}(f_M^*) = \mathbb{E}_p\left[\mathbf{1}\left[f_M^*(x_1, x_2) = y\right]\right]$ denote the accuracy of the Bayes' optimal multimodal model $f_M^*$ (i.e., $P_{acc}(f_M^*) \geq P_{acc}(f_M')$ for all $f_M' \in \mathcal{F}_M$). We have that*

$$2^{I_p(\{X_1,X_2\};Y)-H(Y)} \leq P_{acc}(f_M^*) \leq \frac{I_p(\{X_1,X_2\};Y)+1}{\log|\mathcal{Y}|}, \tag{52}$$

where we can plug in $R + U_1, U_2 + \underline{S} \leq I_p(\{X_1, X_2\}; Y) \leq R + U_1, U_2 + \overline{S}$ to obtain lower $\underline{P}_{acc}(f_M^*)$ and upper $\overline{P}_{acc}(f_M^*)$ bounds on optimal multimodal performance.

*Proof.* We use the bound from Feder & Merhav (1994), where we define the Bayes' optimal classifier $f_M^*$ is the one where given $x_1, x_2$ outputs $y$ such that $p(Y = y|x_1, x_2)$ is maximized over all $y \in \mathcal{Y}$.

Table 6: Full list of best unimodal performance $P_{\text{acc}}(f_i)$, best simple fusion $P_{\text{acc}}(f_{M\text{simple}})$, and best complex fusion $P_{\text{acc}}(f_{M\text{complex}})$ as obtained from the most recent state-of-the-art models. We also include our estimated bounds $(\underline{P}_{\text{acc}}(f_M^*), \overline{P}_{\text{acc}}(f_M^*))$ on optimal multimodal performance.

| | MOSEI | UR-FUNNY | MOSI |
|---|---|---|---|
| $\overline{P}_{\text{acc}}(f_M^*)$ | 1.07 | 1.21 | 1.29 |
| $P_{\text{acc}}(f_{M\text{complex}})$ | 0.88 (Hu et al., 2022) | 0.77 (Hasan et al., 2021) | 0.86 (Hu et al., 2022) |
| $P_{\text{acc}}(f_{M\text{simple}})$ | 0.85 (Rahman et al., 2020) | 0.76 (Hasan et al., 2021) | 0.84 (Rahman et al., 2020) |
| $P_{\text{acc}}(f_i)$ | 0.82 (Delbrouck et al., 2020) | 0.74 (Hasan et al., 2021) | 0.83 (Yang et al., 2020) |
| $\underline{P}_{\text{acc}}(f_M^*)$ | 0.52 | 0.58 | 0.62 |

| | MUSTARD | MIMIC | ENRICO |
|---|---|---|---|
| $\overline{P}_{\text{acc}}(f_M^*)$ | 1.63 | 1.27 | 0.88 |
| $P_{\text{acc}}(f_{M\text{complex}})$ | 0.79 (Hasan et al., 2021) | 0.92 (Liang et al., 2021) | 0.51 (Liang et al., 2021) |
| $P_{\text{acc}}(f_{M\text{simple}})$ | 0.74 (Pramanick et al., 2021) | 0.92 (Liang et al., 2021) | 0.49 (Liang et al., 2021) |
| $P_{\text{acc}}(f_i)$ | 0.74 (Hasan et al., 2021) | 0.92 (Liang et al., 2021) | 0.47 (Liang et al., 2021) |
| $\underline{P}_{\text{acc}}(f_M^*)$ | 0.78 | 0.76 | 0.48 |

The probability that this classifier succeeds is $\max_y p(Y = y | x_1, x_2)$, which is $2^{-H_\infty(Y | X_1 = x_1, X_2 = x_2)}$ where $-H_\infty(Y | X_1, X_2)$ is the min-entropy of the random variable $Y$ conditioned on $X_1, X_2$. Over all inputs $(x_1, x_2)$, the probability of accuracy is

$$P_{\text{acc}}(f_M^*) = \mathbb{E}_{x_1, x_2} \left[ 2^{-H_\infty(Y | X_1 = x_1, X_2 = x_2)} \right] \geq 2^{-\mathbb{E}_{x_1, x_2}[H_\infty(Y | X_1 = x_1, X_2 = x_2)]} \tag{53}$$

$$\geq 2^{-\mathbb{E}_{x_1, x_2}[H_p(Y | X_1 = x_1, X_2 = x_2)]} \geq 2^{-H_p(Y | X_1, X_2)} = 2^{I_p(\{X_1, X_2\}; Y) - H(Y)}. \tag{54}$$

The upper bound is based on Fano's inequality (Fano, 1968). Starting with $H_p(Y | X_1, X_2) \leq H(P_{\text{err}}) + P_{\text{err}}(\log |\mathcal{Y}| - 1)$ and assuming that $Y$ is uniform over $|\mathcal{Y}|$, we rearrange the inequality to obtain

$$P_{\text{acc}}(f_M^*) \leq \frac{H(Y) - H_p(Y | X_1, X_2) + \log 2}{\log |\mathcal{Y}|} = \frac{I_p(\{X_1, X_2\}; Y) + 1}{\log |\mathcal{Y}|}. \tag{55}$$

$\square$

Finally, we summarize estimated multimodal performance as the average between estimated lower and upper bounds on performance: $\hat{P}_M = (\underline{P}_{\text{acc}}(f_M^*) + \overline{P}_{\text{acc}}(f_M^*))/2$.

**Unimodal and multimodal performance**: Table 6 summarizes all final performance results for each dataset, spanning unimodal models and simple or complex multimodal fusion paradigms, where each type of model is represented by the most recent state-of-the-art method found in the literature.

## E  SELF-SUPERVISED MULTIMODAL LEARNING VIA DISAGREEMENT

Finally, we highlight an application of our analysis towards self-supervised pre-training, which is generally performed by encouraging agreement as a pre-training signal on large-scale unlabeled data (Radford et al., 2021; Singh et al., 2022) before supervised fine-tuning (Oord et al., 2018). However, our results suggest that there are regimes where disagreement can lead to synergy that may otherwise be ignored when only training for agreement. We therefore design a new family of self-supervised learning objectives that capture *disagreement* on unlabeled multimodal data.

### E.1  METHOD

We build upon masked prediction that is popular in self-supervised pre-training: given multimodal data of the form $(x_1, x_2) \sim p(x_1, x_2)$ (e.g., $x_1 = $ caption and $x_2 = $ image), first mask out some words $(x_1')$ before using the remaining words $(x_1 \backslash x_1')$ to predict the masked words via learning $f_\theta(x_1' | x_1 \backslash x_1')$, as well as the image $x_2$ to predict the masked words via learning $f_\theta(x_1' | x_2)$ (Singh et al., 2022; Zellers et al., 2022). In other words, maximizing agreement between $f_\theta(x_1' | x_1 \backslash x_1')$ and $f_\theta(x_1' | x_2)$ in predicting $x_1'$:

$$\mathcal{L}_{\text{agree}} = d(f_\theta(x_1' | x_1 \backslash x_1'), x_1') + d(f_\theta(x_1' | x_2), x_1') \tag{56}$$

for a distance $d$ such as cross-entropy loss for discrete word tokens. To account for disagreement, *we allow predictions on the masked tokens $x_1'$ from two different modalities $i, j$ to disagree by a slack*

*variable* $\lambda_{ij}$. We modify the objective such that each term only incurs a loss penalty if each distance $d(x,y)$ is larger than $\lambda$ as measured by a margin distance $d_\lambda(x,y) = \max(0, d(x,y) - \lambda)$:

$$\mathcal{L}_{\text{disagree}} = \mathcal{L}_{\text{agree}} + \sum_{1 \le i < j \le 2} d_{\lambda_{ij}}(f_\theta(x'_1|x_i), f_\theta(x'_1|x_j)) \tag{57}$$

These $\lambda$ terms are hyperparameters, quantifying the amount of disagreement we tolerate between each pair of modalities during cross-modal masked pretraining ($\lambda = 0$ recovers full agreement). We show this visually in Figure 9 by applying it to masked pre-training on text, video, and audio using MERLOT Reserve (Zellers et al., 2022), and also apply it to FLAVA (Singh et al., 2022) for images and text experiments (see extensions to 3 modalities and details in Appendix E).

### E.2 TRAINING DETAILS

We continuously pretrain and then finetune a pretrained MERLOT Reserve Base model on the datasets with a batch size of 8. The continuous pretraining procedure is similar to Contrastive Span Training, with the difference that we add extra loss terms that correspond to modality disagreement. The pretraining procedure of MERLOT Reserve minimizes a sum of 3 component losses,

$$\mathcal{L} = \mathcal{L}_{\text{text}} + \mathcal{L}_{\text{audio}} + \mathcal{L}_{\text{frame}} \tag{58}$$

where each of the component losses is a contrastive objective. Each of the objectives aims to match an independent encoding of masked tokens of the corresponding modality with the output of a Joint Encoder, which takes as

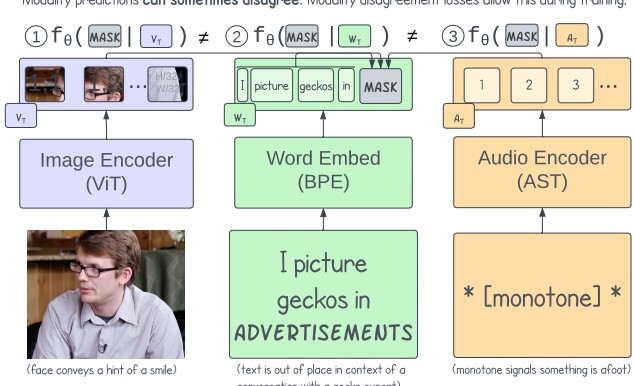

Figure 9: Masked predictions do not always agree across modalities, as shown in this example from the Social-IQ dataset. Adding a slack term enabling pre-training with modality disagreement yields strong performance improvement over baselines.

input the other modalities and, possibly, unmasked tokens of the target modality.

We modify the procedure by adding disagreement losses between modalities to the objective. This is done by replacing the tokens of a modality with padding tokens before passing them to the Joint Encoder, and then calculating the disagreement between representations obtained when replacing different modalities. For example, $\mathcal{L}_{\text{frame}}$ uses a representation of video frames found by passing audio and text into the Joint Encoder. Excluding one of the modalities and passing the other one into the Encoder separately leads to two different representations, $\hat{\mathbf{f}}_\mathbf{t}$ for prediction using only text and $\hat{\mathbf{f}}_\mathbf{a}$ for prediction using only audio. The distance between the representations is added to the loss. Thus, the modified component loss is

$$\mathcal{L}_{\text{disagreement, frame}} = \mathcal{L}_{\text{frame}} + d_{\lambda_{\text{text, audio}}}(\hat{\mathbf{f}}_\mathbf{t}, \hat{\mathbf{f}}_\mathbf{a}) \tag{59}$$

where $d_{\lambda_{\text{text, audio}}}(\mathbf{x}, \mathbf{y}) = \max(0, d(\mathbf{x}, \mathbf{y}) - \lambda_{\text{text, audio}})$, and $d(\mathbf{x}, \mathbf{y})$ is the cosine difference:

$$d(\mathbf{x}, \mathbf{y}) = 1 - \frac{\mathbf{x} \cdot \mathbf{y}}{|\mathbf{x}||\mathbf{y}|} \tag{60}$$

Similarly, we modify the other component losses by removing one modality at a time, and obtain the new training objective

$$\mathcal{L}_{\text{disagreement}} = \mathcal{L}_{\text{disagreement, text}} + \mathcal{L}_{\text{disagreement, audio}} + \mathcal{L}_{\text{disagreement, frame}} \tag{61}$$

During pretraining, we train the model for 960 steps with a learning rate of 0.0001, and no warm-up steps, and use the defaults for other hyperparameters. For every dataset, we fix two of $\{\lambda_{\text{text, audio}}, \lambda_{\text{vision, audio}}, \lambda_{\text{text, vision}}\}$ to be $+\infty$ and change the third one, which characterizes the most meaningful disagreement. This allows us to reduce the number of masked modalities required from 3 to 2 and thus reduce the memory overhead of the method. For SOCIAL-IQ, we set $\lambda_{\text{text, vision}}$ to be 0. For UR-FUNNY, we set $\lambda_{\text{text, vision}}$ to be 0.5. For MUSTARD, we set $\lambda_{\text{vision, audio}}$ to be 0. All training is done on TPU v2-8 accelerators, with continuous pretraining taking 30 minutes and using up to 9GB of memory.

Table 7: Allowing for disagreement during self-supervised masked pre-training yields performance improvements on these datasets. Over 10 runs, improvements that are statistically significant are shown in bold ($p < 0.05$).

| | SOCIAL-IQ | UR-FUNNY | MUSTARD | CARTOON | TVQA |
|---|---|---|---|---|---|
| FLAVA, MERLOT Reserve | $70.6 \pm 0.6$ | $80.0 \pm 0.7$ | $77.4 \pm 0.8$ | $38.6 \pm 0.6$ | $82.0$ |
| + disagreement | $\mathbf{71.1 \pm 0.5}$ | $\mathbf{80.7 \pm 0.5}$ | $\mathbf{78.1 \pm 1.1}$ | $39.3 \pm 0.5$ | $83.0$ |

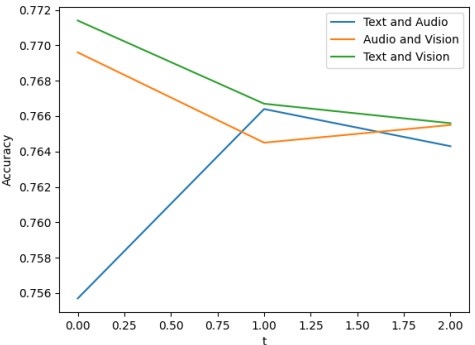

Figure 10: Impact of modality disagreement on model performance. Lower $t$ means we train with higher disagreement between modalities: we find that disagreement between text and vision, as well as audio and vision, are more helpful during self-supervised masking with performance improvements, whereas disagreement between text and audio is less suitable and can even hurt performance.

### E.3 SETUP

We choose four settings with natural disagreement: (1) UR-FUNNY: humor detection from $16,000$ TED talk videos (Hasan et al., 2019), (2) MUSTARD: 690 videos for sarcasm detection from TV shows (Castro et al., 2019), (3) SOCIAL IQ: $1,250$ multi-party videos testing social intelligence knowledge (Zadeh et al., 2019), (4) CARTOON: matching $704$ cartoon images and captions (Hessel et al., 2022), and (5) TVQA: a large-scale video QA dataset based on 6 popular TV shows (Friends, The Big Bang Theory, How I Met Your Mother, House M.D., Grey's Anatomy, Castle) with $122,000$ QA pairs from $21,800$ video clips (Lei et al., 2018).

### E.4 RESULTS

From Table 7, allowing for disagreement yields improvements on these datasets, with those on SOCIAL IQ, UR-FUNNY, MUSTARD being statistically significant (p-value < 0.05 over 10 runs). By analyzing the value of $\lambda$ resulting in the best validation performance through hyperparameter search, we can analyze when disagreement helps for which datasets, datapoints, and modalities. On a dataset level, we find that disagreement helps for video/audio and video/text, improving accuracy by up to 0.6% but hurts for text/audio, decreasing the accuracy by up to 1%. This is in line with intuition, where spoken text is transcribed directly from audio for these monologue and dialog videos, but video can have vastly different information. In addition, we find more disagreement between text/audio for SOCIAL IQ, which we believe is because it comes from natural videos while the others are scripted TV shows with more agreement between speakers and transcripts. Finally, we also scaled to TVQA by incorporating the disagreement objective on top of MERLOT Reserve (Zellers et al., 2022). Unfortunately, running multiple times was not possible due to the large size of the dataset, but our preliminary results show that adding disagreement improves performance from 82% to 83% as compared to the original agreement-based Contrastive Span pretraining (Zellers et al., 2022). We will continue to investigate disagreement-based SSL in large-scale settings in future work.

### E.5 DATASET LEVEL ANALYSIS

We now study the impact of pairwise modality disagreement on the entire dataset on model performance by fixing two modalities $M_1, M_2$ and a threshold $t$, and setting the modality pair-specific disagreement slack terms $\lambda$ according to the rule

$$\lambda_{a,b} = \begin{cases} t, & a = M_1, b = M_2 \\ +\infty, & \text{else} \end{cases}$$

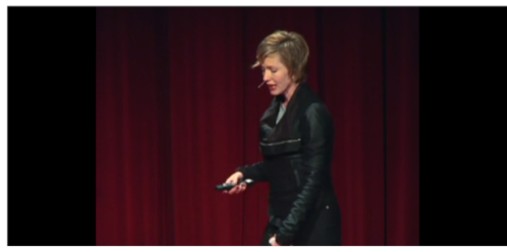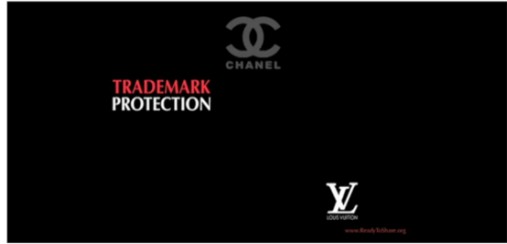

they have trademark protection but no copyright protection ... all they have really is trademark protection

High disagreement                    Low disagreement

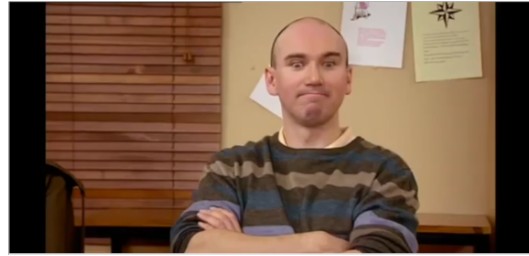

we're all really really looking forward to hearing your story

High disagreement

Figure 11: Examples of disagreement due to uniqueness (up) and synergy (down)

This allows us to isolate $d_{\lambda_{M_1,M_2}}$ while ensuring that the other disagreement loss terms are 0. By decreasing $t$, we encourage higher disagreement between the target modalities. In Figure 10, we plot the relationship between model accuracy and $t$ for the MUStARD dataset to visualize how pairwise disagreement between modalities impacts model performance.

### E.6 DATAPOINT LEVEL ANALYSIS

Finally, we visualize the individual datapoints where modeling disagreement helps in model predictions. After continuously pretraining the model, we fix a pair of modalities (text and video) and find the disagreement in these modalities for each datapoint. We show examples of disagreement due to uniqueness and synergy in Figure 11. The first example is from UR-FUNNY dataset: the moments when the camera jumps from the speaker to their presentation slides are followed by an increase in agreement since the video aligns better with the speech. In the second example on MUStARD, we observe disagreement between vision and text when the speaker's face expresses the sarcastic nature of a phrase. This changes the meaning of the phrase, which cannot be inferred from text only, and leads to synergy.

