# OpenReview forum: "Multimodal Learning Without Labeled Multimodal Data: Guarantees and Applications"
_ICLR.cc/2024/Conference — ICLR 2024 poster_

### Official Review · Reviewer_rHJs · 2023-10-28

**Soundness:** 3 good
**Presentation:** 3 good
**Contribution:** 3 good
**Rating:** 8
**Confidence:** 4

**Summary:**

This study develop a system of information-theoretic definitions and theories that quantify multimodal interaction in a semi-supervised setting. The semi-supervised setting being investigated contains only labelled unimodal datasets and unlabelled co-occurring multimodal data. The authors derive 2 lower bounds and 1 upper bound for Synergy information - emerged information when all modalities incorporate. Additionally, the research demonstrates how these theoretical results can be applied to estimate multimodal model performance, guide data collection, and select appropriate multimodal models for various tasks.

**Strengths:**

Contribution:

- The definitions are set out in accordance with real scenarios and datasets (e.g. MIMIC, MUSTARD,…) and in accordance with previous literatures, which increase the credit and sensibility of the definitions.
- The line of theorem following and based on those definitions are set out reasonable and come with rigorous proof in Appendix.
- By extracting 4 information factors given pairwise marginal distributions, the authors derive the bound of accuracy for optimal multimodal model. This can be used to analyse whether to collect more data and choose a different fusion approach

Prepresentation:

- The authors provides trackable notation system and use those notations throughout the manuscript, which increase its consistency and coherence.
- Clear instructions via reference system and details in Appendix, make it more readable and trackable (e.g. we refer reader to …).

**Weaknesses:**

- Upon deriving lower bound using uniqueness, the reliance on optimal unimodal classifiers seem not practical in most scenarios.
- Upon deriving the upper bound, the relaxation using min-entropy coupling problem might not produce a tight upper boundary.
- While the empirical result support the correct boundary of *Synergy*, there is no investigation on how the bound is closed to *Synergy*, or what is the gap can be determined as close enough.

**Questions:**

How to determine the closeness of the derived boundary?

---

> ### Author Response · Authors · 2023-11-20
> **Rebuttal by Authors**
>
> Thank you for your valuable feedback and insightful comments! We respond to some concerns below:
>
> [optimal unimodal classifiers] During the rebuttal period, we also added experiments ablating the quality of unimodal classifiers by adding label noise (from no noise to very noisy) to make unimodal predictors non-optimal. The bounds are quite robust to these imperfections, still giving close trends of real synergy (see Figure 8, Section C.4 in updated submission pdf). This implies that even imperfect unimodal classifiers can be practically useful. In real-world experiments, the bounds are accurate as shown in the close tracking of synergy on 10 real-world datasets from Table 1 and on 100,000 synthetic datasets in Figure 2.
>
> [upper bound min-entropy coupling] We prove that our upper bounds are **at most a one-bit overapproximation** to the true synergy (Theorem 4, details in Appendix B.6) despite the optimization problem being **intractable** (Theorem 3). It is possible to give a better guarantee by applying very recent advances from the field of minimum-entropy coupling [1], but finding the tightest possible upper bound is an active area of research with implications beyond multimodal learning. **Our upper bound results will improve as new minimum-entropy coupling algorithms are developed.**
>
> What is crucial is that **our bounds are non-vacuous and practically useful** as evidenced by our experiments and their implications on estimating multimodal performance, deciding when to collect multimodal data, and designing multimodal fusion models.
>
> [1] Compton et al., Minimum-entropy coupling approximation guarantees beyond the majorization barrier. AISTATS 2023
>
> [how close the bounds are] As above, we prove that our upper bounds are at most a one-bit overapproximation to the true synergy. **The lower bounds are empirically tighter, for half the points, lower bound 2 is within 0.14 of S and lower bound 1 is within 0.2 of S.**
>
> [what is close enough] A guideline for a bound being ‘close enough’ if it accurately identifies the **maximum interaction** - often the exact value of synergy is not the most important (e.g, whether S is 0.5 or 0.6) but rather synergy relative to other interactions (e.g., if we estimate S in [0.2, 0.6], and exactly compute R = U1 = U2 = 0.1, then we know for sure that S is the most important interaction and can collect data and design multimodal models based on that). We therefore add a new metric measuring how often the maximum interaction stays the same when S is known exactly versus when we estimate a range for S via lower and upper bounds. **We found that our synergy bounds accurately identifies the same highest interaction on all 10 real-world datasets as the true synergy does**. In other words, the estimated synergy bounds correlate exactly with true synergy. For example, the estimated average synergy (average = lower bound + upper bound/2) is as high as 1.05 on ENRICO (true S = 1.02) and as low as 0.21 on MIMIC (true S = 0.02).

---

> > ### Comment · Reviewer_rHJs · 2023-11-21
> > **Final comment of Reviewer rHJs**
> >
> > Dear Authors,
> >
> > Thank you for the further clarification.
> > I have no additional concern for now.
> > Good work.
> >
> >
> > Best,

---

### Official Review · Reviewer_o1cs · 2023-10-31

**Soundness:** 3 good
**Presentation:** 3 good
**Contribution:** 3 good
**Rating:** 6
**Confidence:** 3

**Summary:**

This paper provides two lower bounds and one upper bound to quantify the interactions between modalities for the proposed multimodal semi-supervised setting. One lower bound is based on mutual information between modalities and the other is based on disagreement between separately trained unimodal classifiers. The upper bound is approximated by min-entropy coupling methods. The performances of the bounds to track true interactions on synthetic and real-world datasets are evaluated by experimental results. Examples are shown to use the bounds to guide modality or model selection in multimodal learning.

**Strengths:**

1. The performances of the lower bounds are tight, which demonstrates the effectiveness.
2. The proposed bounds can be used for modality or model selection, which are helpful for multimodal learning.

**Weaknesses:**

1. The performance of the proposed upper bound is not very ideal, more explanations can be provided.
2. No related work is compared in the experimental results. The superiority of this work is not demonstrated.

**Questions:**

See the weaknesses.

---

> ### Author Response · Authors · 2023-11-20
> **Rebuttal by Authors**
>
> Thank you for your valuable feedback and insightful comments! We respond to some concerns below:
>
> [upper bound performance] We prove that our upper bounds are **at most a one-bit overapproximation** to the true synergy (Theorem 4, details in Appendix B.6) despite the optimization problem being intractable (Theorem 3). It is possible to give a better guarantee by applying very recent advances from the field of minimum-entropy coupling [1], but to the best of our knowledge finding the tightest possible upper bound is an active area of research with implications beyond multimodal learning. **Our upper bound results will improve as new minimum-entropy coupling algorithms are developed. The lower bounds are empirically tighter, for half the points, lower bound 2 is within 0.14 of S and lower bound 1 is within 0.2 of S.**
>
> What is crucial is that **our bounds are non-vacuous and practically useful** as evidenced by our experiments and their implications on estimating multimodal performance, deciding when to collect multimodal data, and designing multimodal fusion models.
>
> [1] Compton et al., Minimum-entropy coupling approximation guarantees beyond the majorization barrier. AISTATS 2023
>
> [related work comparisons] **To compare, we implemented 3 alternative definitions for multimodal interactions based on related work, including the original called I-min, WMS, and CI (see [1] for a review) as well as 3 popular non-information theory based methods shapley values, integrated gradients, and CCA**. We note that these definitions need fully paired (x1,x2,y) data, so it is not clear how to estimate them in a semi-supervised settings, so we can only compare in the supervised (x1,x2,y) case. Given supervised data, we add their results to Table 5 in Appendix C.3 of the updated pdf, and summarize main findings here:
>
> 1. I-min [1] can sometimes overestimate synergy and uniqueness. It overestimates synergy on R-only, U1-only, and U2-only datasets, and assigns non-zero unique interactions (U1=0.08, U2=0.07 and U1=0.03, U2=0.04) even though there is no uniqueness.
> 2. WMS [1] is actually synergy minus redundancy, so when they are of equal magnitude WMS cancels out to be 0, leading one to erroneously conclude there is no S or R. For example, it is unable to detect high synergy for S-only data and assigns uniqueness instead. This term can also be negative (S=-0.11 for R-only data).
> 3. CI [1] can also be negative (U1=-0.09, U2=-0.10 for R-only data), and sometimes incorrectly concludes highest uniqueness (U1=0.13, U2=0.14) for S-only data.
> 4. Shapley [2] is based on interactions captured by a trained multimodal model, which means it is unable to guide data collection and model selection, which are key contributions we make. Furthermore, Shapley can only capture overall independence, not separate uniqueness for each modality, it overestimates synergy=0.5 for pure redundancy data and overestimates independence=1.0 for pure synergy data, and estimating Shapley for large, high-dimensional datasets is still an open question.
> 5. Integrated gradients (IG) [3] only gives unimodal contributions of a trained model, which means it is unable to guide data collection and model selection. Moreover, it is not clear how to estimate synergy with IG.
> 6. CCA [4] only gives shared information between modalities, which can be used as a proxy of redundancy, but does not give other interactions. Furthermore, one can learn representations to maximize correlation arbitrarily high (e.g., rho=1.0 for MOSEI), even though actual redundancy is bounded.
>
> **We are not aware of other related work in mathematically formalizing multimodal interactions, much less for semi-supervised data. Most of our results are new theoretical and empirical insights about different multimodal interactions. The mathematical setup we choose based on [5] is well-justified by giving more accurate estimates, and gives the benefit of estimating interactions in the semi-supervised case.**
>
> [1] Griffith and Koch. Quantifying synergistic mutual information. arXiv 2014
>
> [2] Ittner et al., Feature synergy, redundancy, and independence in global model explanations using shap vector decomposition. arXiv 2021
>
> [3] Sundararajan et al., Axiomatic attribution for deep networks. ICML 2017.
>
> [4] Andrew. Deep canonical correlation analysis. ICML 2013
>
> [5] Bertschinger et al., Quantifying unique information. Entropy 2014

---

### Official Review · Reviewer_dPyL · 2023-11-01

**Soundness:** 3 good
**Presentation:** 3 good
**Contribution:** 3 good
**Rating:** 6
**Confidence:** 4

**Summary:**

This paper explores the challenge of understanding how different modalities combine to provide new task-relevant information in a semi-supervised multimodal learning setting. The authors derive lower and upper bounds to quantify the amount of multimodal interactions, with the lower bounds based on shared information and disagreement between unimodal classifiers, and the upper bound derived through connections to min-entropy couplings. The authors also propose a practical algorithm to estimate the lower bounds in practice. The theoretical results in this paper could be used to guide data collection or select appropriate multimodal models for a specific task. Overall, this paper provides a valuable contribution to the field of multimodal learning by providing a framework for quantifying interactions between modalities.

**Strengths:**

1. The paper provides a novel contribution to the field of multimodal learning by deriving lower and upper bounds to quantify the amount of multimodal interactions in a semi-supervised setting. This provides a framework for understanding how different modalities combine to provide new task-relevant information.

2. The paper is theoretically rigorous, with the authors providing a detailed information-theoretic analysis of the problem. The authors also propose a practical algorithm to estimate the lower bounds in practice. The authors show that synergy bounds can be used to predict the performance of multimodal models on held-out data, and to identify modalities that are most important for synergy. This information can be used to guide data collection efforts and improve the performance of multimodal models.

3. The theoretical results in this paper could be used to guide data collection or select appropriate multimodal models for a specific task. This has important practical implications for the development of multimodal learning systems in a variety of domains.

**Weaknesses:**

1. The synergy bounds are only approximate. The authors acknowledge that the synergy bounds are not tight, and that they may underestimate or overestimate the true amount of synergy in a dataset. This is a limitation of any information-theoretic approach to synergy quantification. While the paper provides a theoretical framework for quantifying interactions between modalities, there is limited empirical evaluation of the proposed approach. The authors only provide a proof-of-concept experiment on a small dataset, which may limit the generalizability of the results.

2. The synergy bounds are sensitive to the choice of auxiliary distributions. The authors use a set of auxiliary distributions to compute their lower bound on synergy. The choice of these auxiliary distributions can affect the tightness of the bound. The authors do not provide any guidance on how to choose the auxiliary distributions, which leaves this as a practical challenge for users of the proposed method.

3. The synergy bounds are not applicable to all types of synergy. The authors focus on quantifying synergy that arises from the disagreement between unimodal predictors. However, there are other types of synergy, such as synergy that arises from the complementary strengths of different modalities. The proposed synergy bounds are not applicable to these other types of synergy.

4. The paper does not provide any theoretical guarantees on the performance of multimodal models trained using synergy bounds. The authors show that synergy bounds can be used to predict the performance of multimodal models on held-out data. However, they do not provide any theoretical guarantees on the accuracy of these predictions. This is a limitation of the paper, as it would be useful to know how accurately synergy bounds can predict the performance of multimodal models in practice.

**Questions:**

1. In section 3, what is D_M? Did you consider number of modalities over 2?

2. In Table 3, can you give more details of multimodal or unimodal, and what are the evaluation metrics to define the best models?

3. In section 4.2 RQ1, what performance to estimate? I think authors could have provided more explanations or examples to make the paper more accessible to a wider audience.

4. The authors mentioned observing only labeled unimodal data and some unlabeled multimodal data, did authors conduct experiments on these kind of data or any ablation studies on incompleteness of unimodal data?

---

> ### Author Response · Authors · 2023-11-20
> **Rebuttal by Authors**
>
> Thank you for your valuable feedback and insightful comments! We respond to some concerns below:
>
> [approximate bounds] Since we are operating in a semi-supervised learning setting (Section 2.1), samples from the true joint distribution of $(X_1, X_2, Y)$ are not available: we only have access to **unlabeled** unimodal datasets of the form $(X_1, Y)$ and $(X_2, Y)$ as well as multimodal $(X_1, X_2)$. **It is unreasonable, and in fact, impossible to determine the true synergy** without additional structural assumptions or priors on their multimodal joint distribution $p(x_1, x_2, y)$. There could be two distributions, each satisfying the pairwise marginals $p(x_1, y), p(x_2, y)$ and $p(x_1, x_2)$ but with wildly different joint distributions and synergy. **It is therefore by necessity that we are only able to provide bounds (or approximates), which can overestimate or underestimate synergy.**
>
> [tightness] We actually prove that our upper bounds are **at most a one-bit overapproximation** to the true synergy (Theorem 4, details in Appendix B.6) despite the optimization problem being intractable (Theorem 3). It is possible to give a better guarantee by applying very recent advances from the field of minimum-entropy coupling, but to the best of our knowledge finding the tightest possible upper bound is an active area of research with implications beyond multimodal learning. **Our upper bound results will improve as new minimum-entropy coupling algorithms are developed. The lower bounds are empirically tighter, for half the points, lower bound 2 is within 0.14 of S and lower bound 1 is within 0.2 of S.**
>
> What is crucial is that **our bounds are non-vacuous and practically useful** as evidenced by our experiments and their implications on estimating multimodal performance, deciding when to collect multimodal data, and designing multimodal fusion models.
>
> [1] Compton et al., Minimum-entropy coupling approximation guarantees beyond the majorization barrier. AISTATS 2023
>
> [empirical evaluation] We would like to clarify that our previous experiments are conducted on 100,000 synthetic distributions and on 6 real-world multimodal datasets (MOSEI, UR-FUNNY, MOSI, MUSTARD, MIMIC, ENRICO). The real-world datasets have a total size of 23,000 + 16,000 + 2,199 + 690 + 36,212 + 1,460 = 79,561 datapoints. **During the rebuttal period, we added 4 more real-world multimodal datasets, which now bring the total to more than 700,000 datapoints across 10 real-world datasets.**
> 1. NYCaps [1]: New York Times cartoon images and humorous captions describing these images. 1,820 datapoints.
> 2. IRFL [2]: Images and figurative language (e.g, ‘the car is as fast as a cheetah’ describing an image with a fast car in it). 6,697 datapoints.
> 3. VQA [3]: Question answering about natural images. 614,000 datapoints.
> 4. ScienceQA [4]: Question answering about science problems with scientific diagrams. 21,000 datapoints.
>
> **We added these results to expanded Table 1, which now includes 10 datasets. Our methods are able to lower and upper synergy accurately, both in identifying cases of low synergy and cases of high synergy:**
> 1. NYCaps: true synergy is low at 0.09, estimated average synergy is 0.34
> 2. IRFL: true synergy is lowest at 0, estimated average synergy is 0
> 3. VQA: true synergy is highest at 0.05 (other interactions are 0), estimated average synergy is 0.48
> 4. ScienceQA: true synergy is highest at 0.16, estimated average synergy is 0.84
>
> Adding the new experiments now brings the total to 10 real-world datasets with a total size of 723,078 samples, which we believe is sufficient to prove the generalizability of results.
>
> [1] Hessel et al., Do Androids Laugh at Electric Sheep? Humor "Understanding" Benchmarks from The New Yorker Caption Contest. ACL 2023
>
> [2] Yosef et al., IRFL: Image Recognition of Figurative Language. arXiv 2023
>
> [3] Antol et al., VQA: Visual Question Answering. ICCV 2015
>
> [4] Lu et al., Learn to Explain: Multimodal Reasoning via Thought Chains for Science Question Answering. NeurIPS 2022

---

> ### Author Response · Authors · 2023-11-20
> **Rebuttal by Authors (part 2)**
>
> [auxiliary distributions] We apologize for the confusion - these auxiliary distributions are not something we can ‘choose’, but rather the broader set of multimodal distributions r that match both unimodal marginals $r(x_i,y) = p(x_i,y)$ and modality marginals $r(x_1,x_2) = p(x_1,x_2)$. By optimizing over r, we find distributions that give either less or more synergy in line with the original distribution, which yields appropriate lower and upper bounds.
>
> **There is nothing to choose in the auxiliary distributions, our code runs off-the-shelf without any hyperparameter turning, and is extremely efficient: the computation takes < 1 minute and < 180 MB memory space on average for our large datasets (1,000-10,000 datapoints), which is 3x less time and 15x less memory than training even the smallest multimodal model.**
>
> [Types of synergy] **We did exactly study the 2 types of synergy you mentioned**. In Section 3 we categorize synergy into:
> 1. **Synergy due to the complementary strengths of different modalities, section on ‘synergy and redundancy’, and derive our first lower bound using redundancy (Theorem 1).**
> 2. **Synergy due to disagreement between unimodal predictors, section on ‘synergy and uniqueness, and derive our second lower bound using uniqueness (Theorem 2)**.
> **Figure 1 gives an overview of these 2 types of synergy.**
>
> [theoretical guarantees] **Theorem 5 is exactly a theoretical guarantee on the performance of multimodal models using the estimated synergy bounds**. Specifically, we can convert lower and upper bounds on synergy to lower and upper bounds on the performance of multimodal models. We verify these guarantees in Table 3, where we compare estimated performance with the real-world performance of many multimodal models on 6 real-world datasets. We find that the theoretical equation well matches the real-world performances.
>
> [Section 3] D_M represents the unlabeled multimodal data (x1,x2). D_1 represents labeled modality 1 data (x1,y) and D_2 represents labeled modality 2 data (x2,y).
>
> [>2 modalities] We only considered 2 modalities in this work. Extensions to more than 2 modalities can be challenging due to a larger number of interactions (e.g., x1 and x2 may exhibit synergy, but not with x3); future work can leverage advances in information theory: e.g. one could consider decomposing features hierarchically, or approximating via pairwise synergies.
>
> [Table 3] Unimodal models are the standard single-modality encoders used in each task (e.g, language-only models, vision-only models). Multimodal models represent any model that takes as input both modalities, learns some fused representation, and uses that to predict the label.
>
> Each dataset has their own metrics, which are all classification accuracies over the label space. The best unimodal or multimodal model is the one that achieves the highest classification accuracy on the test set.
>
> The full list of models that obtain the best performance is listed in Table 6, and all the best-performing multimodal models are recent state-of-the-art approaches based on multimodal deep learning. Our estimated lower and upper bounds for multimodal performance very well track the actual performances of unimodal and multimodal models on these real datasets.
>
> [Section 4.2 RQ1] We are given as input some unlabeled multimodal data (x1,x2) and some labeled single-modality data (x1,y) and (x2,y). Using these 3 datasets, the goal is to estimate the performance (accuracy) of a multimodal model f that is trained on labeled multimodal data (x1,x2,y). We show that we can estimate this accuracy even without data (x1,x2,y) and training the model on (x1,x2,y).
> 1. In RQ2, we start from (x1,x2), (x1,y), and (x2,y), estimate accuracy, and show that it helps you decide whether to collect the full (x1,x2,y) or not.
> 2. In RQ3, we start from (x1,x2), (x1,y), and (x2,y), estimate accuracy, and show that it helps you decide what model to use - whether unimodal models, simple multimodal models, or more complex multimodal models.
>
> [labeled unimodal and unlabeled multimodal data] **All our experiments are conducted using access to only unlabeled multimodal data (x1,x2) and some labeled single-modality data (x1,y) and (x2,y).**
> 1. Given only (x1,x2), (x1,y), and (x2,y), our experiments in section 4.1 estimate lower and upper bounds for S, and compare these bounds to the true S estimated from full (x1,x2,y).
> 2. Given only (x1,x2), (x1,y), and (x2,y), our experiments in section 4.2 estimate the lower and upper  bounds on the performance (accuracy) of the multimodal model f trained on the full (x1,x2,y). We show that this gives useful insights for what data to collect and what model to use.

---

> > ### Comment · Reviewer_dPyL · 2023-11-22
> >
> > Thanks for detailed explanation. I think the authors addressed most of my concerns, so I raise my rating to marginally above.

---

### Author Response · Authors · 2023-11-20
**Changes made during rebuttal period**

Dear all reviewers, we are extremely grateful for your valuable feedback and insightful comments. We are glad that you agree that our results are a novel contribution, theoretically rigorous, and has important practical implications for the development of multimodal learning systems in a variety of domains. Your concrete suggestions are a valuable step in this direction, and we have revised our submission accordingly to take these into account. In this short note, we summarize the main changes we made to our submission:

[larger datasets] We added experiments on 4 more real-world multimodal datasets in Table 1, which now bring the total experiments to more than 700,000 datapoints across 10 real-world datasets and 100,000 controlled synthetic distributions. The 4 new datasets include
1. IRFL: Images and figurative language (e.g, ‘the car is as fast as a cheetah’ describing an image with a fast car in it).
2. NYTcartoon: New York Times cartoon images and humorous captions describing these images.
3. VQA: question answering about natural images.
4. ScienceQA: question answering about science problems with scientific diagrams.

Results are consistent and closely track true synergy on all datasets.

[other methods] We implemented 3 other information theoretic measures I-min, WMS, and CI and 3 popular non-info theory measures Shapley values, Integrated gradients (IG), and CCA, adding results to Table 5 of the paper. The mathematical setup we choose is well-justified by giving more accurate estimates, and gives the benefit of estimating interactions in the semi-supervised case.

[tightness] We prove that our upper bounds are at most a one-bit overapproximation to the true synergy. The lower bounds are empirically tighter, for half the points, lower bound 2 is within 0.14 of S and lower bound 1 is within 0.2 of S. We also add a guideline in page 7 of the submission: bounds can be considered ‘close enough’ if it accurately identifies the maximum interaction, which is the most important for data collection and modeling. Our bounds accurately identifies the highest interaction on all 10 real-world datasets as true synergy does.

[imperfect unimodal classifiers] We added experiments ablating the quality of unimodal classifiers by adding label noise to make unimodal predictors non-optimal. The bounds are quite robust to these imperfections, still giving close trends of real synergy (see Figure 8 in submission). Real-world imperfect unimodal classifiers are practically useful for tracking synergy on 10 real-world datasets.

---

### Meta-Review · Area_Chair_cAxD · 2023-12-08

**Metareview:**

The authors present lower and upper bounds to characterise how different modalities contribute to task-relevant interactions in semi-supervised settings using disagreements with unimodal models (lower bounds) and entropies (upper bounds). Practical implications may for example include better informed selection of modalities or multimodal models.

**Justification For Why Not Higher Score:**

This is a solid paper but none of the reviewers is really enthusiastic about it.

**Justification For Why Not Lower Score:**

Most of the concerns of the reviews were addressed in the rebuttal.

---

### Decision · Program_Chairs · 2024-01-16

Accept (poster)